# The Land Use Model Intercomparison Project (LUMIP) contribution to CMIP6: Rationale and experimental design

David M. Lawrence[1], George C. Hurtt[2], Almut Arneth[3], Victor Brovkin[4], Kate V. Calvin[5], Andrew D. Jones[6], Chris D. Jones[7], Peter J. Lawrence[1], Nathalie de Noblet-Ducoudré[8], Julia Pongratz[4], Sonia I. Seneviratne[9], and Elena Shevliakova[10]

[1] National Center for Atmospheric Research, Boulder, CO, USA

[2] University of Maryland, College Park, MD, USA

[3] Karlsruhe Institute of Technology, Garmisch-Partenkirchen, Germany

[4] Max Planck Institute for Meteorology, Hamburg, Germany

[5] Joint Global Change Research Institute, College Park, MD, USA

[6] Lawrence Berkeley National Laboratory, Berkeley, CA, USA

[7] Met Office Hadley Centre, Exeter, UK

[8] Laboratoire des Sciences du Climat et de l'Environnement, Gif-sur-Yvette, France

[9] Institute for Atmospheric and Climate Science, ETH Zurich, Zurich, Switzerland

[10] NOAA/GFDL and Princeton University, Princeton, New Jersey, USA

**Abstract**

Human land-use activities have resulted in large changes to the Earth surface, with resulting implications for climate. In the future, land-use activities are likely to expand and intensify further to meet growing demands for
food, fiber, and energy.  The Land Use Model Intercomparison Project (LUMIP) aims to further advance understanding of the impacts of land-use and land-cover change (LULCC) on climate, specifically addressing the questions: (1) What are the effects of LULCC on climate and biogeochemical cycling (past-future)?  (2) What are the impacts of land management on surface fluxes of carbon, water, and energy and are there regional land-management strategies with promise to help mitigate against climate change?  In addressing these questions,
LUMIP will also address a range of more detailed science questions to get at process-level attribution, uncertainty, data requirements, and other related issues in more depth and sophistication than possible in a multi-model context to date. There will be particular focus on the separation and quantification of the effects on climate from LULCC relative to all forcings, separation of biogeochemical from biogeophysical effects of land-use, the unique impacts of land-cover change versus land management change, modulation of land-use impact on climate by land-
atmosphere coupling strength, and the extent that impacts of enhanced $CO_2$ concentrations on plant photosynthesis are modulated by past and future land use.

LUMIP involves three major sets of science activities: (1) development of an updated and expanded historical and future land-use dataset, (2) an experimental protocol for specific LUMIP experiments for CMIP6, and (3) definition of metrics and diagnostic protocols that quantify model performance, and related sensitivities, with respect to
LULCC.  In this manuscript, we describe the LUMIP activity (2), i.e., the LUMIP simulations that will formally be part of CMIP6.  These experiments are explicitly designed to be complementary to simulations requested in the CMIP6 DECK and historical simulations and other CMIP6 MIPs including ScenarioMIP, C4MIP, LS3MIP, and DAMIP.  LUMIP includes a two-phase experimental design.  Phase one features idealized coupled and land-only model simulations designed to advance process-level understanding of LULCC impacts on climate, as well as to quantify model
sensitivity to potential land-cover and land-use change. Phase two experiments focus on quantification of the historic impact of land use and the potential for future land management decisions to aid in mitigation of climate change. This paper documents these simulations in detail, explains their rationale, outlines plans for analysis, and describes a new subgrid land-use tile data request for selected variables (reporting model output data separately for primary and secondary land, crops, pasture, and urban land-use types). It is essential that modeling groups
participating in LUMIP adhere to the experimental design as closely as possible and clearly report how the model experiments were executed.

*Keywords: Land-use change, climate and Earth system modeling, CMIP6*

## 1. Introduction

Historic land-cover and land-use change has dramatically altered the character of the Earth's surface, directly impacting climate and perturbing natural biogeochemical cycles. Land-use activities are expected to expand and/or intensify in the future to meet increasing human demands for food, fiber, and energy. From a broad perspective, the biogeophysical impacts of land-use and land-cover change (LULCC) on climate are relatively well-understood with observational and modeling studies tending to agree that deforestation has and will lead to cooling in high latitudes and warming in the tropics with more uncertain changes in the mid-latitudes (e.g., Bonan 2008; Davin and de Noblet-Ducoudré 2010; Lee et al. 2011; Li et al. 2016; Pielke et al. 2011; Swann et al. 2012). The impact of land-cover change on, for example global mean surface air temperature, has been and is projected to continue to be relatively small (Brovkin et al. 2013; Lawrence et al. 2012), but, regionally, climate change due to deforestation can be as large or larger than that resulting from increases in greenhouse gas emissions (de Noblet-Ducoudré et al. 2012). Nonetheless, substantial disagreement exists across models in terms of their simulated regional climate response to LULCC (Kumar et al. 2013; Pitman et al. 2009), and some observed effects do not appear to be captured by models (Lejeune et al. 2016), contributing to a lack of confidence in model projections of regional climate change. Variation among future scenarios of land-use change, which could depart significantly from historical trends due to large-scale adoption of either afforestation or biofuel policies, introduces another source of uncertainty that has not been examined in a systematic fashion (Jones et al. 2013b).

The biogeochemical impact of LULCC relates to emissions of greenhouse gases (GHGs) such as $CO_2$, $CH_4$, and $N_2O$ in response to LULCC (e.g., Canadell et al. 2007; Houghton 2003; Pongratz et al. 2009; Shevliakova et al. 2009). Models estimate that the net LULCC carbon flux - the $CO_2$ exchange between vegetation and atmosphere due to LULCC such as emissions due to forest clearing and carbon uptake in regrowth of harvested forest - has accounted for ~25% of the historic increase in atmospheric carbon dioxide concentration (Ciais et al. 2014), but the LULCC flux remains one of the most uncertain terms in the global carbon budget (Houghton et al. 2012). As on the biogeophysical side, models show a wide range of estimates for historic and future emissions due to LULCC (Arora and Boer 2010; Boysen et al. 2014; Brovkin et al. 2013). When emissions of all GHG species due to LULCC are considered, the forcing due to LULCC accounts for ~45% of the total historic (1850 to 2010) changes in radiative forcing (Ward et al. 2014).

At the same time, there is growing awareness that the details of land use matter and that land management or land-use intensification can have as much of an impact on climate as land-cover change itself. Luyssaert et al. (2014) emphasize that while humans have instigated land-cover change over about 18-29% of the ice-free land surface, a much larger fraction of the planet (42-58%) has not experienced land-cover change per se, but is nonetheless managed, sometimes intensively, to satisfy human demands for food and fiber. Furthermore, the temperature impacts, assessed through remote sensing and paired tower sites, are roughly equivalent for land-management change and land-cover change. Other examples of research indicating the importance of specific

aspects of land management are numerous.  For example, irrigation, which has increased substantially over the 20[th] century (Jensen et al. 1990), can directly impact local and regional climate (Boucher et al. 2004; Sacks et al. 2009; Wei et al. 2013). In some regions, cooling trends associated with irrigation area expansion have likely offset warming due to greenhouse gas increases (Lobell et al. 2008a).  Explicit representation of the crop life cycle also appears to be important; Levis et al. (2012) showed that including an interactive crop model into a global climate model (GCM) can improve the seasonality of surface turbulent fluxes and net ecosystem exchange and thereby directly impact weather and climate and the carbon cycle.  In another study, Pugh et al. (2015) found that accounting for harvest, grazing, and tillage resulted in cumulative post-1850 land-use related carbon loss that was 70% greater than in simulations ignoring these processes. There is a hypothesis that increasing crop production over the 20[th] century could account for ~25% of the observed increase in the amplitude of the $CO_2$ annual cycle (Gray et al. 2014; Zeng et al. 2014). Furthermore, agricultural practices can mitigate heat extremes through the cooling effects of irrigation (Lobell et al. 2008b), due to enhanced evapotranspiration associated with cropland intensification (Mueller et al. 2015), or by increasing surface albedo by transitioning to no-till farming (Davin et al. 2014).  Forest management and the harvesting of trees for wood products or fuel is also important and has substantial carbon cycle consequences (Hurtt et al. 2011) with the carbon flux due to wood harvest amounting to an equivalent of up to 15% of the forest net primary production in strongly managed regions such as Europe (Luyssaert et al. 2010). Awareness that land management can impact climate has led to open questions about whether or not there is potential for implementation of specific land management as a tool for local or global climate mitigation (e.g., Canadell and Raupach 2008; Marland et al. 2003).

Due to the predicted increases in global population and affluence as well as the increasing importance of bioenergy, demand for food and fiber is likely to surge during the coming decades. Expansion of active management into relatively untouched regions could satisfy a portion of the growing demand for food and fiber but intensification is likely to play a stronger role in strategies for global sustainability (Foley et al. 2011; Reid et al. 2010). Therefore, we can anticipate a growing contribution from land-management change to the overall impacts of LULCC on the climate system.  The requirement of negative emissions to achieve low radiative forcing targets highlights the need for more comprehensive understanding of the impacts (e.g., on land use, water, nutrients, albedo) and sustainability of carbon removal strategies such as bioenergy carbon capture and storage (BECCS, Smith et al. 2016).

Clearly, the impacts of land cover and land use on climate are myriad and diverse and, while uncertain, are sufficiently large and complex to warrant an expanded activity focused on land-use within CMIP6.  The Land Use Model Intercomparison Project (LUMIP, https://cmip.ucar.edu/lumip) addresses this topic in the context of CMIP6 (Eyring et al. 2015). The goal of LUMIP is to enable, coordinate, and ultimately address the most important science questions related to the effects of land-use on climate. LUMIP scientific priorities and model experiments have been developed in consultation with several existing model intercomparison activities and research programs that focus on the role of land use in climate including the Land-Use and Climate, IDentification of robust impacts project

(LUCID, de Noblet-Ducoudré et al. 2012; Pitman et al. 2009), the Land-use change: assessing the net climate forcing, and options for climate change mitigation and adaptation project (LUC4C, http://luc4c.eu/), the trends in net land carbon exchange project (TRENDY, http://dgvm.ceh.ac.uk/node/9), and the Global Soil Wetness Project (GSWP3). In addition, the LUMIP experimental design is complementary with and in some cases requires simulations from several other CMIP6 MIPs including ScenarioMIP (O'Neill 2016), C4MIP (Jones et al. 2016), LS3MIP (Van den Hurk et al. 2016), DAMIP (Gillett 2016), and RFMIP (Pincus 2016). In all cases, the LUMIP experiments are complementary and not duplicative with experiments requested in these other MIPs. We will reference these cross-MIP interactions throughout this manuscript, where applicable.

### 1.1 LUMIP Activities

The main science questions that will be addressed by LUMIP, in the context of CMIP6 are:

- What are the global and regional effects of land-use and land-cover change on climate and biogeochemical cycling (past-future)?
- What are the impacts of land management on surface fluxes of carbon, water, and energy
- Are there regional land-use or land-management strategies with promise to help mitigate against climate change?

In addressing these questions, LUMIP will also address a range of more detailed science questions to get at process level attribution, uncertainty, data requirements, and other related issues in more depth and sophistication than possible in a multi-model context to date. There will be particular focus on (1) the separation and quantification of the effects on climate from LULCC relative to all forcings, (2) separation of biogeochemical from biogeophysical effects of land-use, (3) the unique impacts of land-cover change versus land-use change, (4) modulation of land-use impact on climate by land-atmosphere coupling strength, and (5) the extent that the direct effects of higher $CO_2$ concentrations on increases in global plant productivity are modulated by past and future land use.

Three major sets of science activities are planned within LUMIP. First, a new set of global gridded land-use forcing datasets has been developed to link historical land-use data and future projections in a standard format required by climate models (Figure 1). This new generation of "land-use harmonization" (LUH2) builds upon past work from CMIP5 (Hurtt et al. 2011), and includes updated inputs, higher spatial resolution, more detailed land-use transitions, and the addition of important agricultural management layers. The new dataset includes annual land-use states, transitions, and management layers for the years 850 to 2100 at 0.25$^\circ$ spatial resolution. Note that land-cover data and forest/non-forest data, as well as land-use transitions, will be provided in the new dataset in order to help minimize misinterpretation of the land-use dataset that occurred in CMIP5 where, for example, the strong afforestation in RCP4.5 was not captured in Community Earth System Model (CESM) simulations because of differing assumptions embedded within the CESM land use translator (a software package that translates the LUH

data into CESM land-cover datasets) and the LUH dataset (Di Vittorio et al. 2014). Several harmonized future land-use trajectories will be processed for the period 2016-2100 in support of the ScenarioMIP Shared Socioeconomic Pathway scenarios (see Section 2.3.2).  Cropland is disaggregated into five crop functional types based on input

data from FAO and Monfreda et al. (2008).  Crop rotations are also included. Grazing lands are disaggregated into managed pastures and rangelands based on input data from the updated HYDE3.2 dataset (updated from HYDE3.1, Klein Goldewijk et al. 2011), which also provides inputs for gridded cropland, urban, and irrigated area. The modeling process includes new underlying maps of potential biomass density and biomass recovery rate, which are used to disaggregate both primary and secondary natural vegetation into forested and non-forested land. It also

includes a new representation of shifting cultivation rates and extent, constrains forest loss between the years 2000-2012 with Landsat-based forest loss data from Hansen et al. (2013), and uses a new historical wood harvest reconstruction based on updated FAO data, new HYDE population data, and other sources. The LUH2 dataset will include several new agricultural management layers such as gridded nitrogen fertilizer usage based on Zhang et al. (2015), gridded irrigated areas (based on HYDE3.2), and gridded areas flooded for rice (also based

on HYDE3.2), as well as the disaggregation of wood harvest into fuelwood and industrial roundwood (i.e., timber that is cut for uses other than for fuel). Future scenarios (years 2016-2100) will also include biofuel management layers.  To help address the issue of sensitivity to uncertainty in historical land-use forcing, two alternative historical land-use reconstructions have also been developed. These alternatives are based on same data sources, use same algorithms,  and are provided in same format as the reference LUH2 product, but  span

range of uncertainty in the key historical input datasets for agriculture and wood harvest. Specifically, the 'high' reconstruction, assumes high historical estimates for crop and pasture and wood harvest, and the 'low' reference assumes low estimates for each of these terms, relative to the reference.

The LUH2 dataset is available through the LUMIP website (https://cmip.ucar.edu/lumip) and will be described in a separate publication in this CMIP6 Special Issue.  Guidance on use of the data will be provided in the LUH2 dataset

paper and through the LUMIP website.

Second, an efficient model experiment design, including both idealized and scenario-based cases, is defined that will enable isolation and quantification of land-use effects on climate and the carbon cycle (see Section 2). The LUMIP experimental protocol enables integrated analysis of coupled and land-only (forced with observed meteorology) models which will support understanding and assessment of the forced response and climate

feedbacks associated with land use and the relationship of these responses to land and atmosphere model biases.

Third, a set of metrics and diagnostic protocols will be developed to quantify model performance, and related sensitivities, with respect to land use (see Section 3). De Noblet-Ducoudré et al (2012) identified the lack of consistent evaluation of a land model's ability to represent a response to a perturbation such as land-use change as a key contributor to the large spread in simulated land-cover change responses seen in LUCID. As part of this

activity, benchmarking data products will be identified to help constrain models. Where applicable, these metrics

will be incorporated into land model metrics packages such as the International Land Model Benchmarking (ILAMB, http://www.ilamb.org/) system.

New output data standardization will also enrich and expand analysis of model experiment results. Particular emphasis within LUMIP is on archival of subgrid land information in CMIP6 experiments (including LUMIP experiments and other relevant experiments from ScenarioMIP, C4MIP, and the CMIP historical simulation). In most land models, physical, ecological, and biogeochemical land state and surface flux variables are calculated separately for several different land surface type or land management 'tiles' (e.g., natural and secondary vegetation, crops, pasture, urban, lake, glacier). Frequently, including in the CMIP5 archive, tile-specific quantities are averaged and only grid-cell mean values are reported. Consequently, a large amount of valuable information is lost with respect to how each land-use type responds to and interacts with climate change and direct anthropogenic modifications of the land surface. LUMIP has developed a protocol and associated data request for CMIP6 for selected key variables on separate land-use tiles within each grid cell (primary and secondary land, crops, pastureland, urban; see Section 4).

**1.2 Relevance of LUMIP to CMIP6 questions and WCRP Grand Challenges**

Land-use change is an essential forcing of the Earth System, and as such LUMIP is directly relevant and necessary for CMIP6 Question 1 (Eyring et al. 2015): "How does the Earth System respond to forcing?". LUMIP will also play a strong role in addressing the WCRP Grand Challenges (GC), particularly with respect to GC7 "determining how biogeochemical cycles and feedbacks control greenhouse gas concentrations and climate change," GC3 "understanding the factors that control water availability over land", and GC4 "assessing climate extremes, what controls them, how they have changed in the past and how they might change in the future." Due to the broad range of effects of land-use change and the major activities proposed, LUMIP is also of cross-cutting relevance to CMIP6 science questions 2 "What are the origins and consequences of systematic model biases?" and 3 "How can we assess future climate change given climate variability, climate predictability, and uncertainties in scenarios?"

**1.3 Definitions of land cover, land use, and land management**

Within LUMIP, we rely on prior definitions of land cover, land use, and land management (Lambin et al. 2006). Land cover refers to "the attributes of the Earth's land surface and immediate subsurface, including biota, soil, topography, surface and groundwater, and human (mainly built-up) structures", and is represented in land models by categories like forest, grassland, cropland or urban areas. Land use is the "purpose for which humans exploit the land cover"; e.g., a grassland may be left in its natural state, mowed, or utilized as rangeland for livestock. Land management refers to ways in which humans treat vegetation, soil, and water, and is captured in land models by processes such as irrigation, use of fertilizers and pesticides, crop species selection, or methods of wood harvesting (selective logging versus clear cutting). Thus, within the same land-cover category several land uses can occur, and

within the same land-use category, management practices can differ. Land-cover change usually goes in hand with land-use change, but the opposite is not true. Land-cover change can also be driven by natural processes such as a change of the biogeographic vegetation distribution due to climate shifts or natural disturbance (Davies-Barnard et al. 2015; Schneck et al. 2013).  For the purposes of LUMIP, the term "LULCC" includes anthropogenically-driven land-cover change only.

## 2. Experimental design and description

In this section, we begin with a discussion and recommendations on the specification of land use in CMIP6 Diagnostic, Evaluation and Characterization of Klima (DECK) and historical experiments and other MIP experiments (Section 2.1).   Also in this section, we outline the full set of requested LUMIP experiments (Sections 2.2 and 2.3). LUMIP includes a two phase, tiered, model experiment plan. Phase one features a coupled model simulation with an idealized deforestation scenario that is designed to advance process-level understanding and to quantify model sensitivity to land-cover change impacts on climate and biogeochemical stocks and fluxes.  Phase one also includes a factorial set of land-only model simulations that allow assessment of the forced-response of land-atmosphere fluxes to land-cover change as well as examination of the impacts of various land use and land-management practices. Phase two experiments will focus on the quantification of the historic impact of land use and the potential for future land management decisions to aid in the mitigation of climate change.  A forum for discussion of the experiments and for distribution of minor updates or clarifications to the experimental design will be hosted at the LUMIP website (https://cmip.ucar.edu/lumip).

Details of the model experiments are described below.  The full set of LUMIP experiments include:

- Tier 1 (high priority): 500 years GCM/ESM; ~650  years land-only
- Tier 2 (medium priority): 500 years GCM/ESM; up to 1500 to 3000 years land-only

Note that these totals only represent the LUMIP-sponsored simulations.  LUMIP analysis requires control simulations from other MIPs, e.g., a pre-industrial control DECK simulation or a CMIP6 historical simulation.  We note the required 'parent' simulation and responsible MIP, where applicable.

In Sections 2.2 and 2.3, we describe each experiment in detail.  Also included is the scientific rationale for the particular experiment or set of experiments.  The heading for each experiment includes several relevant pieces of information according to the following format - **Short description** (*CMIP6 experiment ID*, model configuration, Tier X, # years) - where the model configuration is either land-only (offline land simulations forced with observed meteorology), GCM (fully coupled simulation, concentration-driven), or ESM (fully coupled simulation, emissions-driven).

### 2.1 Land-use treatment in the CMIP6 DECK, historical experiments, and other MIP experiments

There exists a large diversity in representation of LULCC among different land models, and therefore it is typically non-trivial to define what is meant by the terms land use and in particular the term "constant land use." Several CMIP6 simulations both within LUMIP and in other CMIP6 MIPs require land use to be held constant in time including (1) DECK experiments including $CO_2$-concentration and $CO_2$-emission driven pre-industrial control simulations (*piControl*), abrupt quadrupling of $CO_2$ (*abrupt-4×CO2*) and 1%yr$^{-1}$ $CO_2$ increase (*1pctCO2*) simulations, (2) LUMIP no land-use change simulations (Section 2.3.1), (3) C4MIP idealized simulations including biogeochemically-coupled 1%yr$^{-1}$ $CO_2$ increase (*1pctCO2-bgc*) and other C4MIP Tier 2 idealized simulations, and (4) ScenarioMIP extension simulations for the period 2100-2300 (*ssp126-ext, ssp585-ext*) for which land-use data will not be provided.

LUMIP provides the following recommendations to clarify treatment of constant land use. Land cover and land use should be fixed according to the LUH2 specifications for the constant land use reference year (e.g., year 1850 for the DECK pre-industrial control simulation, year 2100 for ScenarioMIP extension simulations). The fraction of cropland and pastureland, as well as the crop type distribution should be held constant. Any land management (e.g., irrigation, fertilization) that exists for the constant land use year should be maintained at the same level. Wood harvesting for timber and shifting cultivation, specified by the LUH2 land-use reconstructions (i.e., through transition matrices or the mass of harvested wood), should be implemented if a model's land-use component permits these processes to be maintained through time at a specified level. If the fire model utilizes population density or other anthropogenic forcings to determine fire ignition and/or suppression rates, then this forcing should also be held constant. We recognize that the diversity of model approaches means that the definition and requirements for constant land management may differ across models. Groups will need to make their own decisions with respect to the treatment of land management in constant land-use scenarios, for example with respect to specification of harvesting on croplands, grazing on pastureland, application of fertilizers, level of irrigation, and wood harvest. Wood harvest, in particular, may require model-specific treatment since turning off wood harvest in the ScenarioMIP 2100-2300 extension runs is likely to result in unrealistic carbon stock trends, while maintaining wood harvest at year 2100 levels for an additional 200 years could unrealistically decimate the forests where the LUH2 datasets indicate wood harvest is happening in 2100. We stress that the individual modeling group decisions should be made within the context of achieving an equilibrated biogeophysical and biogeochemical (e.g., carbon, nitrogen) land state for the pre-industrial 1850 control configurations and to minimize any discontinuities in the shift between a constant land-use simulation and a subsequent transient land-use simulation (see next paragraph for further clarification and discussion). Furthermore, the treatment of constant land use and land management should be clearly documented for each model and experiment. Because some land models are driven by annual maps of land use and others require transition rates between different land-use categories, LUMIP will provide two different 1850 constant land-use datasets – fraction of pastures and crops in 1850 and a one-time set of gross transitions from potential vegetation to the 1850 land-use state.

LUMIP acknowledges and endorses the need for flexible strategies to initialize CMIP6 historical simulations and DECK AMIP simulations. This flexibility is necessitated by (1) considerable structural differences among CMIP6-participating land models, especially with respect to land use (e.g., models with and without wood harvest) and vegetation dynamics (e.g., prescribed versus prognostic vegetation type and age distributions), (2) different spin-up strategies for land-only models versus coupled GCMs and ESMs (e.g., spin-up for potential vegetation versus

constant 1850 land use), and (3) uncertainties in PI-Control experiments due to omission of documented secular multi-century trend in vegetation and soil carbon storage and land-use carbon emissions prior to 1850 (Houghton et al 2010). There are several strategies that have been used in the past and discussed by the modeling groups at present time, including:

- a "seamless" transition from the PI-control to historical as suggested by C4MIP (Jones et al. 2016);

- a "bridge" experiment from an equilibrated ESM spin-up with potential vegetation and subsequent application of land-use scenario applied at a year prior to 1850 (Sentman et al. 2011; Shevliakova et al. 2013).

Consequently, LUMIP does not provide any recommendation on land initialization but requests that all modeling groups document their initialization procedure for their CMIP6 historical simulations and report any differences in

biogeophysical and biogeochemical land states between the 1850 pre-industrial control and the beginning of the CMIP6 historical simulations in 1851. As noted above, a forum for discussion along with additional recommendations and clarifications with respect to initialization, the configuration of 'constant land use', use of the LUH2 data, and other topics will be maintained through the LUMIP website (https://cmip.ucar.edu/lumip).

**2.2. Phase 1 experiments**

Phase 1 consists of two sets of experiments: (a) idealized coupled deforestation experiment that enables analysis of the biogeophysical and biogeochemical response to land-cover change and the associated changes in climate in a controlled and consistent set of simulations (Table 1) and (b) a series of offline land-only simulations to assess how the representation of land cover and land management affects the carbon, water, and energy cycle response to land-use change (Table 2).

**2.2.1    Global deforestation** (*deforest-glob*, GCM, Tier 1, 80 years)

*Description*: Idealized deforestation experiment in which 20 million $km^2$ of forest area (covered by trees) is converted to natural grassland over a period of 50 years with a linear rate of 400,000 $km^2$ $yr^{-1}$, followed by 30 years of constant forest cover (Figure 2A). This simulation should be branched from an 1850 control simulation (*piControl*); all pre-industrial forcings including $CO_2$ concentration and land-use maps and land-management should

be maintained as in the *piControl* and discussed in Section 2.1. The branch should occur at least 80 years prior to the end of the piControl simulation so that *deforest-glob* and *piControl* can be directly compared. In order to concentrate the deforestation from grid cells with predominant forest cover, deforestation should be restricted to

the top 30% of land grid cells in terms of their area of tree cover.  Effectively, this concentrates the deforestation in the tropical rainforest and boreal forest regions (Figure 3).  To do this:

1. Sort land grid cells by forest area and select the top 30% (*gcdef*, Figure 2B).

2. Calculate tree plant type loss for each year at each grid cell by attributing the 400,000 km$^2$ yr$^{-1}$ forest loss proportionally to their forest cover fraction across the *gcdef* grid cells.

Step 2 is formalized as follows. Let *f(x,y,t)* be the forest fraction in grid cell (*x,y*) at the end of year *t* (0 ≤ *t* ≤ 80)*,* *A(x,y)* is the area of the grid cell (million km$^2$). At t=0 (initialization of *deforest-glob*), forest fraction should be equal

to that of the year 1850 in the *piControl*.  The total forest area, F$_{tot}$ (million km$^2$), within the grid cells identified for deforestation (*gcdef*) in Step 1 is:

$$F_{tot} = \sum_{gcdef} f(x,y,t=0)A(x,y) \tag{1}$$

If *F$_{tot}$* is more than 20 million km$^2$, then the scaling coefficient *k$_{gcdef}$* is

$$k_{gcdef} = \frac{20}{F_{tot}} \leq 1 \tag{2}$$

and temporal development of forest fraction in deforested grid cells is calculated as follows:

$$f(x,y,t) = \begin{cases} f(x,y,t=0)(1-\dfrac{k_{gcdef}t}{50}) & 0 < t \leq 50 \\ f(x,y,t=0)(1-k_{gcdef}) & t > 50 \end{cases} \tag{3}$$

If *F$_{tot}$* is less than  or equal to 20 million km$^2$, then the scaling coefficient *k$_{cgef}$* is taken as 1.

Trees should be replaced with natural unmanaged grasslands.  Land use and land management should be maintained at 1850 levels as in the *piControl* experiment.  All above ground biomass (cWood, cLeaf, cMisc) should be removed and below ground biomass (cRoot) transferred to appropriate litter pools (Figure 2C).  If there is no separation of above and below ground biomass in the model, then the whole vegetation biomass pool (cVeg)

should be removed. The replacement of forest with natural grasslands should be done in such a way that the carbon (and nitrogen if applicable) from the forested soil is maintained and allowed to evolve according to natural model processes. If initial forest cover in the *gcdef* grid cells is less than 20 million km$^2$ then should linearly remove all the forested area from the *gcdef* grid cells over 50 years and report the total area of forest removed.  Note that even with substantially different initial forest cover in CCSM4 versus MPI-ESM-P (the examples shown in Figure 3),

the prescribed land-cover change is quite similar for both models when using this deforestation protocol and that modelling groups should strive to produce similar deforestation patterns.

Note that implementation of the deforestation is likely to differ for models with and without vegetation dynamics. Applying deforestation for models without dynamic vegetation should be straightforward as the deforestation can be applied through a time series of land-cover maps that each group can generate. For models with dynamic

vegetation, if possible, vegetation dynamics should be turned off in areas where deforestation is being applied.

Outside the deforested areas, vegetation dynamics can be maintained since the tree cover response to the climate change induced by deforestation is expected to be small over the 80-year simulation time scale.

We recognize that each participating land model has its own unique structures that may or may not be adequately covered in the above description sketched on the Figure 2. Each modelling group should implement the deforestation in a manner that makes the most sense for their particular modelling system. It is important, however, that all groups strive to produce a spatial and latitudinal deforestation signal that replicates that shown in Figure 3 as closely as possible. The goal of this experiment is to impose deforestation patterns that are as similar as possible across models so as to limit the impact of across-model differences in deforestation patterns on the multi-model evaluation of deforestation impacts on climate and carbon fluxes.

*Rationale*: This experiment is designed to be conceptually analogous to the 1% per year $CO_2$ simulation in the DECK. Prior idealized global or regional deforestation simulations (Badger and Dirmeyer 2015, 2016; Bala et al. 2007; Bathiany et al. 2010; Davin and de Noblet-Ducoudré 2010; Lorenz et al. 2016; Snyder 2010) have proven informative and highlighted how both biogeophysical and biogeochemical forcings due to land-use change contribute to temperature changes, how the ocean can modulate the response, and how remote effects of LULCC can be detected in some situations. However, differences in implementation of realistic historic or projected land-cover change across different models is a problem that has plagued prior land-cover change model intercomparison projects, with a third to a half - depending on season and variable - of the differences in climate response attributable to differences in imposed land cover (Boisier et al. 2012). The relatively simple LUMIP idealized deforestation protocol will enhance uniformity in the prescribed deforestation and therefore enable more direct and meaningful comparison of model responses to deforestation. The gradual deforestation allows a comparison across models with respect to what amplitude of forest loss is needed before a detectable signal emerges at the local and global level, and will provide insight into detection and attribution of land-cover change impacts at regional scales.

**2.2.2 Land-only land-cover and land-use simulations** (*land-xxxx*, land-only; *land-hist*, *land-hist-altStartYear* and *land-noLu* are Tier 1, all others Tier 2, up to 13 simulations, 165 to 315 years each).

*Description*: A set of land-only simulations that are identical to the LS3MIP (Van den Hurk et al. 2016) historical land-only (*land-hist*; Table 2) simulation except with each simulation differing from the *land-hist* simulation in terms of the specific treatment of land use or land management, or in terms of prescribed climate. Note that all simulations should be forced with the default reanalysis dataset provided through LS3MIP (GSWP3 at time of writing). The primary control experiment is *land-hist* this is defined in LS3MIP. This experiment is required (Tier 1), even if the modeling group is not contributing to the full set of LS3MIP experiments. The *land-hist* simulation should include land cover, land use, and land management that is identical to that used in the coupled CMIP6 historical simulation (see next paragraph for more discussion). Two of the LUMIP simulations - *land-hist-*

*altStartYear* and *land-hist-noLu* - are Tier 1. The remaining experiments are Tier 2.  Detailed descriptions of the factorial set of simulations are listed in Table 2.

We anticipate that only a limited number of participating land models will be able to perform all the experiments, but the experimental design allows for models to submit the subset of experiments that are relevant for their model. In some instances, groups may also have a more advanced land model in terms of its representation of land-use-related processes than that which is used in the coupled CMIP6 historical simulation.  In these cases, we request that models submit the LUMIP Tier 1 land-only experiments with the configuration of the land model used in the coupled model CMIP6 historical simulation, but groups are encouraged to provide an additional set of land-only simulations with their more advanced model configuration.

*Rationale*: This factorial series of experiments serves several purposes and is designed to provide a detailed assessment of how the specification of land-cover change and land management affects the carbon, water, and energy cycle response to land-use change. This set of experiments utilizes state-of-the-art land model developments that are planned across several contacted modeling centers and will contribute to the setting of priorities for land use for future CMIP activities. The potential analyses that will be possible through this set of experiments is vast.  We highlight several particular analysis foci here:

(a) The *land-hist* and *land-noLu* simulations will provide context for the global coupled CMIP6 historical simulations, enabling the disentanglement of the LULCC forcing (changes in water, energy and carbon fluxes due to land-use change) from the response (changes in climate variables such as temperature and precipitation that are driven by LULCC-induced surface flux changes), though differences in the coupled model and observed climate forcing will need to be taken into account. The land-only simulations also allow more detailed quantification of the net LULCC flux

(b)  Relative influence of various aspects of land management on the overall impact of land use on water, energy, and carbon fluxes.  For example, comparing the *land-hist* experiment to the experiment with no irrigation (*land-crop-noirrig*) will allow a multi-model assessment of whether or not the increasing use of irrigation during the 20[th] century is likely to have significantly altered trends of regional water and energy fluxes (and therefore climate) or crop yield/carbon storage in agricultural regions.

(c) Pre-industrial land conversion for agriculture was substantial (Pongratz et al. 2008) and has long term and non-negligible legacy effects on the carbon cycle that last well beyond the standard 1850 starting year of CMIP6 historical simulations (Pongratz and Caldeira 2012). By comparing *land-hist* with *land-hist-altStartYear* across a range of models, we can further establish how important pre-1850 land use is for the historical (1850-2005) land carbon stock trajectory.

(d) Gross land-use transitions, especially due to shifting cultivation, can exceed net transitions by a factor of two or more (Hurtt et al. 2011).  Accounting for gross transitions instead of just net transitions results in 15-40% higher simulated net land-use carbon fluxes (Hansis et al. 2015; Stocker et al. 2014; Wilkenskjeld et

al. 2014). For models that can represent shifting cultivation, a parallel experiment (*lnd-hist-noShiftcultivate*) in which shifting cultivation is turned off (net transition) through an alternative set of provided land-use transitions will allow evaluation of the impact of shifting cultivation across a range of models and assumptions (Figure 4).

(e) Comparison of effects of LULCC on surface climate and carbon fluxes (which can be calculated by comparing historical and no-LULCC simulations) between the land-only simulations and the global coupled model simulations (Section 2.3.1) allows assessment of consequences of model climate biases on LULCC effects.

(f) Uncertainty in the land-use history reconstruction is itself a source of uncertainty in the impacts of historic LULCC. The alternative land-use history simulations (*land-hist-altLu1* and *land-hist-altLu2*) in combination with the default land-use history simulation (land-hist) provide information on the sensitivity of the models to a range of plausible reconstructions of land-use history.

*Impact of historic meteorological forcing datasets:* It is critical to acknowledge that all observed historic forcing datasets are subject to considerable errors and uncertainty and that the weather and climate variability and trends represented in these datasets may not accurately reflect reality, especially in remote regions where limited data went into either the underlying reanalysis or the gridded products. These limitations pose a challenge when comparing the model outputs (like latent heat flux, for example) to observed estimates because biases may actually be a function of biases in the meteorological forcing dataset rather than deficiencies in the model. While the land-only LUMIP simulations will only be driven with a single atmospheric forcing dataset (the reference dataset used in the *land-hist* experiment of LS3MIP), the sensitivity of land model output to uncertainty in atmospheric forcing will be assessed in more depth within LS3MIP, which can inform the assessment of the land-only LUMIP simulations.

**2.3. Phase 2 experiments**

The Phase 2 LUMIP experiments are designed to provide a multi-model quantification of the impact of historic LULCC on climate and carbon cycling and to assess the extent to which land management could be utilized as a climate change mitigation tool. This set of experiments includes land-only and coupled historical and future simulations that are derivatives of historical or future simulations within LS3MIP, ScenarioMIP, C4MIP as well as the CMIP6 Historical simulation with land use held constant or modified to an alternative land-use scenario (Table 3). These simulations will be used to assess the role of land use on climate from the perspective of both the biogeophysical and biogeochemical impacts and are likely to be of interest to DAMIP, C4MIP, ScenarioMIP, and LS3MIP.

**2.3.1 Historical no land-use change experiment** (*hist-noLu*; concentration-driven, Tier 1, 165 years)

*Description*: Historical simulation that is identical to CMIP6 historical concentration-driven simulation except that land use is held constant. All land use and management (irrigation, fertilization, wood harvest, gross transitions

exceeding net transitions) is maintained at 1850 levels, in exactly the same way as done for the CMIP6 pre-industrial control simulation (*piControl*).

*Rationale*: This simulation, when compared to the CMIP6 historical simulation, isolates the biogeophysical impact of land-use change on climate and addresses the CMIP6 science question "How does the Earth system respond to forcing?" For models that are run with a diagnostic land carbon cycle, the difference in carbon stocks between *hist-noLu* and the *CMIP6 historical* simulation represents the integrated net LULCC flux. Note that the parallel set of land-only simulations (LS3MIP *land-hist* experiment and LUMIP *land-noLu* experiment, see Sect. 2.1.3) will enable groups to disentangle the contributions of land-use-change induced effects on surface fluxes from atmospheric feedbacks and response (e.g., Chen and Dirmeyer 2016), though the influence of differences in land forcing in coupled versus land-only simulations will need to be taken into account during the analysis. This experiment is directly relevant for detection and attribution studies (DAMIP).

**2.3.2 Future land-use policy sensitivity experiments** (*ssp370-ssp126Lu* and *ssp126-ssp370Lu*, GCM concentration-driven, Tier 1, 2015-2100; *esm-ssp585-ssp126Lu*, ESM emission-driven, Tier 1, 2015-2100)

*Description*: These experiments are derivatives of ScenarioMIP (*ssp370* and *ssp126*, see below for short description of the Shared Socioeconomic Pathways (SSP) land-use scenarios) and C4MIP (*esm-ssp85*) simulations (Figure 5). In each case, the LUMIP experiment is identical to the 'parent' simulation except that an alternative land-use dataset is used. All other forcings are maintained from the parent simulation.

*Rationale*: Both concentration-driven and emission-driven LUMIP alternative land-use simulations are requested. Concentration-driven variants of ScenarioMIP *ssp370* and *ssp126* are required but each using the land-use scenario from the other: i.e., LUMIP simulation *ssp370-ssp126Lu* will run with all forcings identical to *ssp370* except for land use which is to be taken from *ssp126*. These simulations permit analysis of the biogeophysical climate impacts of projected land use and enable preliminary assessment of land use and land management as a regional climate mitigation tool (green arrows on Figure 5). Note that these simulations should be considered sensitivity simulations since they will include a set of forcings that are inconsistent with each other (e.g., land use from SSP1-2.6 in a simulation that in all other respects is equivalent to SSP3-7). This particular set of simulations was selected because the projected land-use trends in SSP3-7 and SSP1-2.6 diverge strongly with SSP3-7 representing a reasonably strong deforestation scenario and SSP1-2.6 including significant afforestation (see Figure 6). These experiments will provide a direct test of an assumption underlying the SSP framework, namely that a particular radiative forcing level can be achieved by multiple socioeconomic scenarios with negligible effect on the resulting climate (Van Vuuren et al. 2014), an assumption that may not hold if patterns of land-use change associated with alternative SSPs diverge significantly enough from one another (Jones et al. 2013b). Furthermore, including experiments in both low and medium/high radiative forcing scenarios allows examination of the extent to which the impact of land-use change differs at different levels of climate change and at different levels of $CO_2$ concentration (red arrows on Figure 5). These sets of experiments can be utilized to provide partial guidance on

the utility of careful land management as a climate mitigation strategy (Canadell and Raupach 2008; Marland et al. 2003).

Emission-driven simulations allow assessment of the full feedback (biogeophysical + biogeochemical) due to land-use change onto climate. In these simulations the ESMs simulate the concentration of atmospheric $CO_2$ in response to prescribed boundary conditions of anthropogenic emissions. Biogeophysical effects operate in the same way as in concentration-driven simulations but in addition, the carbon released or absorbed due to land-use change will affect how the $CO_2$ concentration of the atmosphere evolves in time. Additionally, emission-driven simulations permit assessment of consistency between Integrated Assessment Model predictions (which typically include the biogeochemical effect of land use as a carbon source, but neglect the biophysical effects) about land use and land-use change carbon fluxes with ESM modeled land-use emissions. C4MIP has requested an emission-driven variant to *ssp585,* which will be performed in concentration-driven mode for ScenarioMIP. This will allow quantification of the effects of the climate-carbon cycle feedback on future CO2 and climate change (brown arrow on Figure 5). In LUMIP we request a further SSP5-8.5 simulation: emission-driven but with land use taken from SSP1-2.6. This experiment (*esm-ssp585-ssp126Lu)* will therefore parallel the C4MIP emission-driven experiment (*esm-ssp585)* but will allow us to quantify the full effects of a different land-use scenario through both biophysical and biogeochemical processes (blue arrow on Figure 5).

*Land-use scenarios in SSPs:* The scenarios chosen for use in CMIP6 were developed as part of the Shared Socioeconomic Pathways (SSP) effort (Van Vuuren et al. 2014). Five SSPs were designed to span a range of challenges to mitigation and challenges to adaptation. These SSPs can be combined with RCPs to provide a set of scenarios that span a range of socioeconomic assumptions and radiative forcing levels (Riahi et al. 2016). ScenarioMIP selected eight scenarios from this suite for use in CMIP6. Within LUMIP, we focus on three of these scenarios in our experimental design, chosen because they span a range of future land-use projection (see Popp et al. 2016 for more comprehensive discussion of land-use trajectories). The SSP5-8.5 is a high radiative forcing scenario, reaching 8.5 W m$^{-2}$ in 2100, with relatively little land-use change over the coming century. The increase in radiative forcing is driven by increased use of fossil fuels; however, the combination of a relatively small population and high agricultural yields leads to little expansion of cropland area (Kriegler et al. 2016). In contrast, the SSP3-7 is a world with a large population and limited technological progress, resulting in expanded cropland area (Fujimori et al. 2016). In the SSP1-2.6, efforts are made to limit radiative forcing to 2.6 W m$^{-2}$. These mitigation efforts include reduced deforestation as well as reforestation and afforestation, leading to a scenario where forest cover increases over the coming century (Van Vuuren et al. 2016). Figure 6 shows global time series of forest area, cropland area, pastureland area, wood harvest, area equipped for irrigation, and nitrogen fertilization amounts in the SSP scenarios, highlighting those scenarios selected by ScenarioMIP and LUMIP.

## 3. Land-use metrics and analysis plans

**3.1 Land-use metrics**

A goal of LUMIP is to establish a useful set of model diagnostics that enable a systematic assessment of land use-climate feedbacks and improved attribution of the roles of both land and atmosphere in terms of generating these feedbacks. The need for more systematic assessment of the terrestrial and atmospheric response to land-cover change is one of the major conclusions of the LUCID studies. Boisier et al. (2012) and de Noblet-Ducoudré *et al*

(2012) argue that the different land use-climate relationships displayed across the LUCID models highlights the need to improve diagnostics and metrics for land surface model evaluation in general and the simulated response to LULCC in particular. These sentiments are consistent with recent efforts to improve and systematize land model assessment (e.g., Abramowitz 2012; Best et al. 2015; Kumar et al. 2012; Luo et al. 2012; Randerson et al. 2009). LUMIP will promote a coordinated effort to develop biogeophysical and biogeochemical metrics of model

performance with respect to land-use change that will help constrain model dynamics. These efforts dovetail with expanding emphasis in CMIP6 on model performance metrics. Several recent studies have utilized various methodologies to infer observationally-based historical change in land surface variables impacted by LULCC or divergences in surface response between different land-cover types (Boisier et al. 2013, 2014; Lee et al. 2011; Lejeune et al. 2016; Li et al. 2015; Teuling et al. 2010; Williams et al. 2012).

The availability of both land-only and coupled historic simulations enables a more systematic assessment of the roles of land and atmosphere in the simulated response to land-use change. With both coupled and uncoupled experiments with and without land-use change, we can systematically disentangle the simulated LULCC forcing (changes in land surface water, energy and carbon fluxes due to land-use change) from the response (changes in climate variables such as T and P that are driven by LULCC-driven changes in surface fluxes).

LUMIP also proposes to develop a set of analysis metrics that succinctly quantify a model response to land use across a range of spatial scales and temporal scales that can then be used to quantitatively compare model response across different models, regions, and land management scenarios. For a given variable, say surface air temperature, diagnostic calculations will be completed for a pair of simulations (offline or coupled) with and without land-use change. Across a range of spatial scales, spanning from a single grid cell up to regional,

continental, and global, seasonal mean differences between control and land-use change simulations will be examined. Differences will be expressed, for example, both in terms of seasonal mean differences and in terms of signal to noise (where 'noise' refers to the natural interannual climate variability simulated in the model). Lorenz et al. (2016) emphasize the importance of testing for field significance, especially in the context of evaluating the statistical significance of remote responses to LULCC.

**3.2 Net LULCC carbon flux: loss of additional sink capacity and the net land-use feedback**

To quantify the climatic and carbon cycle consequences of LULCC and land management consistently across models, care has to be taken that the same conceptual framework is applied. Pongratz et al. (2014) have highlighted this issue for the net LULCC carbon flux. The large spread in published estimates of the net LULCC flux

can be substantially attributed to differing definitions that arise from different model and simulation setups. These definitions differ in particular with respect to the inclusion of two processes, the loss of additional sink capacity (LASC) and the land-use carbon feedback. The LASC, which is an indirect LULCC flux, occurs when conversion of land from natural lands (forests) to managed lands (crops or pasture) reduces the capacity of the land biosphere to take up anthropogenic carbon dioxide in the future (e.g., Gitz and Ciais 2003). While small historically it may be of the same order as the net LULCC flux without LASC for future scenarios of strong $CO_2$ increase (Gerber et al. 2013; Mahowald et al. 2016; Pongratz et al. 2014). The land-use carbon feedback can be assessed in emission-driven simulations where LULCC carbon fluxes alter the atmospheric $CO_2$ concentration and the land-use changes also affect the climate through biogeophysical responses, both of which can then feed back onto the productivity of both natural and managed vegetation. Over the historical period, a substantial fraction of the LULCC emissions have been offset with increased vegetation growth. Calculating the net LULCC flux by differencing carbon stocks from a pair of simulations with and without LULCC will lead to net LULCC flux estimates that are about 20-50% lower when calculated from a pair of emission-driven simulations (which include the land use carbon feedback) compared to a pair of land-only simulations (Pongratz et al. 2014; Stocker and Joos 2015).

Within LUMIP, several different model configurations are used that include the LASC and the land-use carbon feedback to different extents (Figure 7). Note that to isolate the effect of LULCC emissions from those of fossil-fuel emissions, a reference simulation is needed, which may be a no-LULCC simulation or a simulation with an alternative LULCC scenario. In the case of the idealized deforestation experiments, where $CO_2$ is kept constant over time, all changes in carbon stocks can be directly attributed to LULCC. The net LULCC flux, as quantified from the land-only simulation, will differ slightly from that calculated in GCM simulations since the GCM simulations include biogeophysical climate feedbacks from LULCC. The difference in net LULCC flux between two LULCC scenarios as derived from the ESM setup follows a different definition, as the land-use carbon feedback is included and its effects cancel only partly by difference of the two simulations.

**3.3 Radiative Forcing**

A recognized limitation within CMIP5 was the difficulty in diagnosis of the radiative forcing due to different forcing mechanisms such as well-mixed GHGs, aerosols or land-use change. In addition, the regionally concentrated nature of biophysical land-use forcing limits the insight gained from quantifying it in terms of a global mean metric (or more strictly the Effective Radiative Forcing, ERF; Davin et al. 2007; Jones et al. 2013a; Myhre et al. 2013). Experiments were performed within CMIP5 to explore different model responses to individual forcings but were not designed to distinguish how each forcing led to a radiative forcing of the climate system versus how the climate system responded to that forcing. For CMIP6, RFMIP is designed to address this gap by including a factorial set of atmosphere-only simulations to diagnose the ERF due to each forcing mechanism individually. Andrews et al. (2016) performed the Radiative Forcing MIP (RFMIP) land use experiment to diagnose the historical ERF from land use in HadGEM2-ES and found a forcing of -0.4 W m$^{-2}$ or about 17% of the total present-day anthropogenic

radiative forcing.  Other studies indicate that the combined radiative forcing effect of land-use change may be as large as ~40% of total present-day anthropogenic radiative forcing, when accounting for emissions of all GHG species due to LULCC (Ward et al. 2014).  LUMIP will benefit from groups performing the RFMIP land-use experiment in addition to the LUMIP simulations.

**3.4 Modulation of land-use change signal by land-atmosphere coupling strength**

An axis of analysis that has not been investigated in great detail is how a particular model's regional land-atmosphere coupling strength signature (Guo et al. 2006; Koster et al. 2004; Seneviratne et al. 2010; Seneviratne et al. 2013) affects simulations of the climate impact of land-use change.   One can hypothesize that LULCC in a region where the land is tightly coupled to the atmosphere, generally due to the presence of a soil moisture-limited evapotranspiration regime (Koster et al. 2004; Seneviratne et al. 2010), will result in a stronger climate response than the same LULCC in a region where the atmosphere is not sensitive to land conditions. In a single model study of Amazonian deforestation, Lorenz and Pitman (2014) find that this is indeed the case – small amounts of deforestation in a part of the Amazon domain where the model simulates strong land-atmosphere coupling has a larger impact on temperature than extensive deforestation in a weakly coupled region.  Similarly, Hirsch et al. (2015) show that different planetary boundary layer schemes, which lead to different land-atmosphere coupling strengths, can modulate the impact of land-use change on regional climate extremes.  LUMIP will collaborate with LS3MIP to systematically investigate the inter-relationships between land-atmosphere coupling strength, which can be diagnosed in any coupled simulation (e.g., Dirmeyer et al. 2014; Seneviratne et al. 2010), and LULCC impacts on climate and establish to what extent differences in land-atmosphere coupling strength across models (Koster et al. 2004) contribute to differences in modeled LULCC impacts.

**3.5 Extremes**

There is evidence that land surface processes strongly affect hot extremes, as well as drought development and heavy precipitation events, in several regions (Davin et al. 2014; Greve et al. 2014; Seneviratne et al. 2010; Seneviratne et al. 2013), and that these relationships could also change with increasing greenhouse gas forcing (Seneviratne et al. 2006; Wilhelm et al. 2015). Therefore, the role of LULCC needs to be better investigated, both in the context of the detection and attribution of past changes in extremes (Christidis et al. 2013) – in coordination with DAMIP – and in assessing its impact on projected changes in climate extremes. In particular, recent studies show that LULCC could affect temperature extremes more strongly than mean temperature, through a combination of changes in albedo (Davin et al. 2014) and accumulated changes in soil moisture content (Wilhelm et al. 2015). Careful assessment will be necessary to validate the inferred relationships between LULCC and extremes, given partly contradicting results with respect to the effects of LULCC on climate extremes in models and observations (Lejeune et al. 2016; Teuling et al. 2010).

### 4. Subgrid data reporting

To address challenges of analyzing effects of LULCC on physical and biogeochemical state of land and its interactions with the atmosphere (e.g., analyses proposed in Section 3.2-3.5), LUMIP is including a Tier 1 data request of sub-grid information *for four sub-grid categories* (i.e., tiles) to permit more detailed analysis of land-use induced surface heterogeneity.  The rationale for this request is that relevant and interesting sub-grid scale data that represents the heterogeneity of the land surface is available from current land models, but is not being used since sub-grid scale quantities are typically averaged to grid cell means prior to delivery to the CMIP database. Several recent studies have demonstrated that valuable insight can be gained through analysis of subgrid information.  For example, Fischer et al. (2012) used sub-grid output to show that not only is heat stress higher in urban areas compared to rural areas in the present-day climate, but also that heat stress is projected to increase more rapidly in urban areas under climate change.  Malyshev et al. (2015) found a much stronger signature of the climate impact of LULCC at the subgrid level (i.e., comparing simulated surface temperatures across different land-use tiles within a grid cell) than is apparent at the grid cell level.  Subgrid analysis can also lead to improved understanding of how models operate.  For example, Schultz et al. (2016) showed, through subgrid analysis of the Community Land Model, that the assumption that plants share a soil column and therefore compete for water and nutrients has the side effect of an effective soil heat transfer between vegetation types which can alias into individual vegetation type surface fluxes.  Furthermore, reporting carbon pools and fluxes by tiles will enable assessment of land-use carbon fluxes not only with the standard method of differencing land-use and no land-use experiments (e.g., as described in Section 3.2) but also within a single land-use experiment, utilizing bookkeeping approaches (Houghton et al. 2012) which allow a more direct comparison of observed and modeled carbon inventory

#### 4.1 Types of land-use tiles

Four land-use categories are requested for selected key variables: (1) primary and secondary land (including bare ground and vegetated wetlands), (2) cropland, (3) pastureland, and (4) urban (Table 4). Other sub-grid categories such as lakes, rivers and glaciers are excluded from this request. The proposed set of land-use sub-grid reporting units closely corresponds to land-use categories to be used in the CMIP6 historical land-use reconstructions and future scenarios. Primary (i.e., natural vegetation never affected by LULCC activity) and secondary land (i.e., natural vegetation that has previously been harvested or abandoned agricultural land with potential to regrow) are combined because most land components of ESMs models do not yet distinguish between these two land types.

#### 4.2 Requested variables and rules for reporting

Overall, there are 5 classes of variables that are requested. These variables describe (a) the subgrid structure and how it evolves through time, (b) biogeochemical fluxes, (c) biogeophysical variables, (d) LULCC fluxes and carbon transfers (Figure 8), and (e) carbon stocks on land-use tiles. A list of requested land-use tile variables is shown in

Table 5.  However, this list is subject to change.  Modelers should refer to the CMIP6 output request documents for the final variable list.

Subgrid tile variables should be submitted according to the following structure, using Leaf Area Index (LAI) as an example: laiLut (lon, lat, time, landusetype4) – where the landusetype4 dimension has an explicit order of psl, crp, pst, urb where "psl" = primary and secondary land, "crp" = cropland, "pst" = pastureland, and "urb" = urban.

It is recognized that different models have very different implementation of LULCC processes and may only be able to report a subset of variables/land-use tiles, but models are requested to report according to the following rules:

- The sum of the fractional areas for psl + crp + pst + urb may not add up to 1 for grid cells with lakes, glaciers or other land sub-grid categories.

- If a model does not represent one of the requested land-use tiles, then it should report for these tiles with missing values.

- In cases where more than one land-use tile shares information then duplicate information should be provided on each tile (e.g., if pastureland and cropland share the same soil then duplicate information for soil variables should be provided on the pst and crp tiles).

- If a model does not represent one of the requested variables for any of the subgrid land-use tiles, then this variable should be omitted.

- Note that for variables where for a particular model the concept of a tiled quantity is not appropriate, that quantity should only be reported at the grid-cell level.  An example is Anthropogenic Product Pools (APP).  Many models do not track APP at the subgrid tile level, instead aggregating all sources of APP into a single grid-cell level APP variable.  In this case, APP should only be reported at the grid cell level as per the CMIP request.

**4.3 Land-use tile-reporting/aggregation for example models**

*Community Land Model (CLM) example*

CLM captures a variety of ecological and hydrological sub-grid characteristics (Figure 9, Lawrence et al. 2011; Oleson et al. 2013).  Spatial land surface heterogeneity in CLM is represented as a nested subgrid hierarchy in which grid cells are composed of multiple land units, snow/soil columns, and PFTs. Each grid cell can have a different number of land units, each land unit can have a different number of columns, and each column can have multiple PFTs. The first subgrid level, the land unit, is intended to capture the broadest spatial patterns of subgrid heterogeneity. The CLM land units are glacier, lake, urban, vegetated, and crop.  The land unit level can be used to further delineate these patterns. For example, the urban land unit is divided into density classes representing the

tall building district, high density, and medium density urban areas. The second subgrid level, the column, is
intended to capture potential variability in the soil and snow state variables within a single land unit. For example, the vegetated land unit could contain several columns with independently evolving vertical profiles of soil water and temperature. Similarly, the crop land unit is divided into multiple columns, two columns for each crop type (irrigated and non-irrigated). The central characteristic of the column subgrid level is that this is where the state variables for water and energy in the soil and snow are defined, as well as the fluxes of these components within
the soil and snow. Regardless of the number and type of plant function types (PFTs) occupying space on the column, the column physics operates with a single set of upper boundary fluxes, as well as a single set of transpiration fluxes from multiple soil levels. These boundary fluxes are weighted averages over all PFTs. Currently, for glacier, lake, and vegetated land units, a single column is assigned to each land unit.

In order to meet requirements of the LUMIP sub-grid reporting request, the following aggregation would be
required for CLM:

- Primary and secondary land (psl): vegetated land unit includes all primary and secondary land which includes all natural vegetation and bare soil
- Crops (crp): crop land unit including all non-irrigated and irrigated crops
- Pastureland: not explicitly treated in CLM, reported as missing value
- Urban (urb): urban land unit including tall building, high density, and medium density areas
- Lakes and glaciers are not included in any of the LUMIP subgrid categories, so are not reported

*GFDL LM3 example*

The GFDL CMIP5 land component LM3 (Shevliakova et al. 2009) resolves sub-grid land heterogeneity with respect to different land-use activities: each grid cell includes up to 15 different tiles (including a bare soil tile) to represent
differences in above- and below-ground hydrological and carbon states (Figure 10). A grid cell could have one cropland tile, one pasture tile, one natural tile, and up to 12 secondary land tiles as well as lake and glacier tiles. Secondary tiles refer to lands that were harvested (i.e., prior primary or secondary) or abandoned agricultural lands, pastures and croplands. The tiling structure of LM3 and ESM2 was designed to work with the CMIP5 LUH dataset (Hurtt et al. 2011). Changes in the area and type of tiles occur annually based on gross transitions from the
LUH dataset. Similarly to the scenario design, secondary or agricultural lands are never allowed to return to primary lands. The physical and ecological states and properties of each of the tiles are different, and the physical and biogeochemical fluxes between land and the atmosphere are calculated separately for every tile. Each cropland, pasture and secondary tile has three anthropogenic pools with three different residence times (1 year, 10 years, and 100 years. For LUMIP sub-grid tile reporting, all secondary and natural tiles will be aggregated into
the primary and secondary tile (PSL). For each requested land-use tile the three different residence-time anthropogenic pools will be aggregated into one.

**5. Summary**

Here, we have outlined the rationale for the Land Use Model Intercomparison Project (LUMIP) of CMIP6. We provided detailed descriptions of the experimental design along with analysis plans and instructions for subgrid
land-use tile data archiving. The efficient, yet comprehensive, experimental design, which has been developed through workshops and discussions among the land-use modeling and related communities over the past two years, includes idealized and realistic experiments in coupled and land-only model configurations. These experiments are designed to advance process-level understanding of land-cover and land-use impacts on climate, to quantify model sensitivity to potential land-cover and land-use change, to assess the historic impact of land use,
and to provide preliminary evaluation of the potential for targeted land use and management as a method to contribute to the mitigation of climate change. In addressing these topics, LUMIP will also study more detailed land-use science questions in more depth and sophistication than possible in a multi-model context to date. Analyses will focus on the separation and quantification of the effects on climate from LULCC relative to all forcings, separation of biogeochemical from biogeophysical effects of land use, the unique impacts of land-cover
change versus land-use change, modulation of land-use impact on climate by land-atmosphere coupling strength, the role of land-use change on climate extremes, and the extent that impacts of enhanced $CO_2$ concentrations on plant photosynthesis are modulated by past and future land use.

**Data availability**

As with all CMIP6-endorsed MIPs the model output from the LUMIP simulations described in this paper will be
distributed through the Earth System Grid Federation (ESGF). The natural and anthropogenic forcing datasets required for the simulations will be described in separate invited contributions to this Special Issue and made available through the ESGF with version control and digital object identifiers (DOI's) assigned. Links to all forcings datasets will be made available via the CMIP Panel website.

**Author contribution**

DML and GCH are co-leads of LUMIP. DML wrote the document with contributions from all other authors.

**Acknowledgements**

We would like to thank Andy Pitman, Paul Dirmeyer, Alan DiVittorio, and Ron Stouffer for their thoughtful and constructive reviews that led to considerable improvements to the document. DML is supported by the US Department of Energy grants DE-FC03-97ER62402/A010 and DE-SC0012972 and US Department of Agriculture
grant 2015-67003-23489. JP is supported by the German Research Foundation's Emmy Noether Program. S.I.S. acknowledges support from the European Research Council (ERC DROUGHT-HEAT project). AA acknowledges support by the EC FP7 project LUC4C (grant no. 603542) and the Helmholtz Association through its ATMO

programme.  NdN acknowledges support by the EC FP7 project LUC4C (grant no. 603542) and by all participants to former LUCID exercises.

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

Table 1: Idealized deforestation experiment designed to gain process understanding and to assess biogeophysical role of land-cover change on climate and inter-compare modeled biogeochemical response to deforestation (concentration-driven).

| Experiment ID | Experiment Name | Experiment Description | Years |
|---|---|---|---|
| **deforest-glob** | Idealized transient global deforestation | Idealized deforestation experiment, 20 million km$^2$ forest removed linearly over a period of 50 years, with an additional 30 years with no specified change in forest cover (Tier 1).This simulation should be branched from an 1850 control simulation (***piControl***)**;** all pre-industrial forcings including CO$_2$ concentration and land-use maps and land management should be maintained as in the ***piControl*** as discussed in Section 2.1 | 80 years |

Table 2. **Land-only** land-cover, land-use, and land-management change simulations. Assess relative impact of land-cover, land-use, and land-management change on fluxes of water, energy, and carbon; forced with historical observed climate. The simulations *land-hist*, *land-hist-altStartYear* and *land-noLu* are Tier 1, all other simulations are Tier 2. All simulations should be pre-industrial to 2015 where pre-industrial start can be either 1850 or 1700 depending on model.

| Experiment ID | Description | Notes |
|---|---|---|
| **land-hist** | Same land model configuration, including representation of land cover, land use, and land management, as used in coupled CMIP6 historical simulation with all applicable land-use features active. Start year either 1850 or 1700 depending on standard practice for particular model. All forcings transient including $CO_2$, N-deposition, aerosol deposition, etc. Shared simulation with LS3MIP. | This simulation can and likely will be a different configuration across models due to different representations of land use for each model. See LS3MIP protocol for full details including details on forcing dataset and spinup |
| **land-hist-altStartYear** | Same as *land-hist* except starting from either 1700 (for models that typically start in 1850) or 1850 (for models that typically start in 1700). | Comparison to *land-hist* indicates impact of pre-1850 land-use change. |
| **land-noLu** | Same as *land-hist* except no land-use change (see Section 2.1 for explanation of no land use). | |
| **land-hist-altLu1** **land-hist-altLu2** | Same as *land-hist* except with two alternative land-use history reconstructions, that span uncertainty in agriculture and wood harvest. Specifically, the *altLu1* is a 'high' reconstruction, assumes high historical estimates for crop and pasture and wood harvest and *altLu2* is a 'low' reference assumes low estimates for each of these terms, relative to the reference dataset. | In combination with *land-hist*, allows assessment of model sensitivity to different assumptions about land-use history reconstructions. Note that land use in 1700 and 1850 will be different to that in *land-hist* so model will need to be spunup again for both alternative datasets. Note that these reconstructions do not span the entire range of uncertainty and the simulations should be considered sensitivity simulations. |
| **land-cCO2** | Same as *land-hist* except with $CO_2$ held constant | |
| **land-cClim** | Same as *land-hist* except with climate held constant | Continue with spinup forcing looping over first 20 years of meteorological forcing data. |
| **land-crop-grass** | Same as *land-hist* but with all new crop and pastureland treated as unmanaged grassland | For this simulation, treat cropland like natural grassland without any crop management in terms of biophysical properties but is treated as agricultural land for dynamic vegetation (i.e. no competition with natural vegetation areas). |
| **land-crop-noIrrigFert** | Same as *land-hist* except with plants in cropland area utilizing at least some form of crop management (e.g., planting and harvesting) rather than simulating cropland vegetation as a natural grassland... Irrigated area and fertilizer area/use should be held constant. | Maintain 1850 irrigated area and fertilizer area/amount and without any additional crop management except planting and harvesting. Irrigation amounts with irrigated area allowed to change. |
| **land-crop-noIrrig** | Same as *land-hist* but with irrigated area held at 1850 levels; only relevant if *land-hist* utilizes at least some form of crop management (e.g., planting and harvesting) | Maintain 1850 irrigated area. Irrigation amounts within the 1850 irrigated area allowed to change |
| **land-crop-noFert** | Same as *land-hist* but with fertilization rates and area held at 1850 levels/distribution; only relevant if *land-hist* utilizes at least some form of crop | |

| | management (e.g., planting and harvesting) | |
|---|---|---|
| **land-noPasture** | Same as *land-hist* but with grazing and other management on pastureland held at 1850 levels/distribution, i.e. all new pastureland is treated as unmanaged grassland (as in *land-crop-grass*). | |
| **land-noWoodHarv** | Same as *land-hist* but with wood harvest maintained at 1850 amounts/areas | Wood harvest due to land deforestation for agriculture should continue yielding non-zero anthropogenic product pools |
| **land-noShiftcultivate** | Same as *land-hist* except shifting cultivation turned off. Only relevant for models where default model treats shifting cultivation (see Figure 4) | An additional LUC transitions dataset will be provided as a data layer within LUMIP LUH2 dataset with shifting cultivation deactivated. |
| **land-noFire** | Same as *land-hist* but with anthropogenic ignition and suppression held to 1850 levels | For example, if ignitions are based on population density, maintain constant population density through simulation |

Table 3: **Coupled Model** Phase 2 simulations, all Tier 1.

| Experiment ID | Experiment Name | Experiment Description | Years |
|---|---|---|---|
| **hist-noLu** | Historical with no land-use change | Same as concentration-driven **CMIP6 historical** (Tier 1) except with LULCC held constant.  See section 2.1 for explanation of no land use. Two additional ensemble members requested in Tier 2. | 1850-2014 |
| **ssp370-ssp126Lu** | SSP3-7 with SSP1-2.6 land use | Same as ScenarioMIP **ssp370** (SSP3-7 deforestation scenario, Tier 1) except use land use from **ssp126** (SSP1-2.6 afforestation scenario); concentration-driven. Two additional ensemble members requested (Tier 2) contingent on ScenarioMIP **ssp370** large ensemble (Tier 2) being completed | 2015-2100 |
| **ssp126-ssp370Lu** | SSP1-2.6 with SSP3-7 land use | Same as ScenarioMIP **ssp126** (SSP1-2.6 afforestation scenario, Tier 1) except use land use from **ssp370** (SSP3-7 deforestation scenario); concentration-driven. | 2015-2100 |
| **esm-ssp585-ssp126Lu** | Emissions-driven SSP5-8.5 with SSP1-2.6 land use | Same as C4MIP **esm-ssp585** (Tier 1) except use SSP1-2.6 land use (afforestation scenario); emission driven | 2015-2100 |

Table 4.  Land-use tile types and abbreviations.

| Land-use Tile Type | Land-Use Tile Abbreviation | Comment |
|---|---|---|
| Primary and secondary land | psl | Forest, grasslands, and bare ground |
| Cropland | crp | |
| Pastureland | pst | Includes managed pastureland and rangeland |
| Urban settlement | urb | |

Table 5.  List of requested variables on land-use tiles.  Note that this list may be updated.  Modelers should refer to the CMIP6 variable request lists for the final list.

| Variable short name | Variable Long Name | Comments |
|---|---|---|
| **Biogeochemical and ecological variables** | | |
| gppLut | gross primary productivity on land use tile | |
| raLut | plant respiration on land use tile | |
| nppLut | net primary productivity on land use tile | |
| cTotFireLut | total carbon loss from natural and managed fire on land use tile, including deforestation fires | Different from LMON this flux should include all fires occurring on the land use tile, including natural, man-made and deforestation fires |
| rhLut | soil heterotrophic respiration on land use tile | |
| necbLut | net rate of C accumulation (or loss) on land use tile | Computed as npp minus heterotrophic respiration minus fire minus C leaching minus harvesting/clearing. Positive rate is into the land, negative rate is from the land.  Do not include fluxes from anthropogenic pools to atmosphere |
| nwdFracLut | fraction of land use tile tile that is non-woody vegetation ( e.g. herbaceous crops) | |
| **Biogeophysical variables** | | |
| tasLut | near-surface air temperature (2m above displacement height, i.e. t_ref) on land use tile | |
| tslsiLut | surface 'skin' temperature on land use tile | temperature at which long-wave radiation emitted |
| hussLut | near-surface specific humidity on land use tile | Normally, the specific humidity should be reported at the 2 meter height |
| hflsLut | latent heat flux on land use tile | |
| hfssLut | sensible heat flux on land use tile | |
| rsusLut | surface upwelling shortwave  on land use tile | |
| rlusLut | surface upwelling longwave on land use tile | |
| sweLut | snow water equivalent on land use tile | |
| laiLut | leaf area index on land use tile | Note that if tile does not model lai, for example, on the urban tile, then should be reported as missing value |
| mrsosLut | Moisture in Upper Portion of Soil Column of land use tile | the mass of water in all phases in a thin surface layer; integrate over uppermost 10cm |
| mrroLut | Total runoff from land use tile | the total runoff (including "drainage" through the base of the soil model) leaving the land use tile portion of the grid cell |
| mrsoLut | Total soil moisture | |
| irrLut | irrigation flux | |
| fahUrb | Anthropogenic heat flux | Anthropogenic heat flux due to human activities such as space heating and cooling or traffic or other energy consumption |
| **LULCC fluxes and carbon transfers** | | |
| fProductDecompLut | flux from anthropogenic pools on land use tile into atmosphere | If a model has separate anthropogenic pools by land use tile |
| fLulccProductLut | carbon harvested due to land-use or land-cover change process that enters anthropogenic product pools on tile | This annual mean flux refers to the transfer of carbon primarily through harvesting land use into anthropogenic product pools, e.g.,deforestation or wood harvesting from primary or secondary lands, food harvesting on croplands, harvesting (grazing) by animals on pastures. |
| fLulccResidueLut | carbon transferred to soil or litter pools due to land-use or land-cover change processes on tile | This annual mean flux due refers to the transfer of carbon into soil or litter pools due to any land use or land-cover change activities |
| fLulccAtmLut | carbon transferred directly to atmosphere due to any land-use or land-cover change activities including deforestation or agricultural fire | This annual mean flux refers to the transfer of carbon directly to the atmosphere due to any land-use or land-cover change activities. |
| **Carbon stock variables** | | |
| cSoilLut | carbon  in soil pool on land use tiles | end of year values (not annual mean) |

| | | |
|---|---|---|
| cVegLut | carbon in vegetation on land use tiles | end of year values (not annual mean) |
| cLitterLut | carbon in above and belowground litter pools on land use tiles | end of year values (not annual mean) |
| cAntLut | anthropogenic pools associated with land use tiles | anthropogenic pools associated with land use tiles into which harvests are deposited before release into atmosphere PLUS any remaining anthropogenic pools that may be associated with lands which were converted into land use tiles during reported period. Does NOT include residue which is deposited into soil or litter; end of year values (not annual mean) |
| **LULCC fraction changes** | | |
| fracLut | fraction of grid cell for each land use tile | end of year values (not annual mean); note that fraction should be reported as fraction of land grid cell |
| fracOutLut | annual gross fraction of land use tile that was transferred into other land use tiles | cumulative annual fractional transitions out of each land use tile; for example, for primary and secondary land use tile, this would include all fractional transitions from primary and secondary land into cropland, pastureland, and urban for the year; note that fraction should be reported as fraction of land grid cell |
| fracInLut | annual gross fraction that was transferred into this tile from other land use tiles | cumulative annual fractional transitions into each land use tile; for example, for primary and secondary land use tile, this would include all fractional transitions from cropland, pastureland, and urban into primary and secondary land over the year; note that fraction should be reported as fraction of land grid cell |

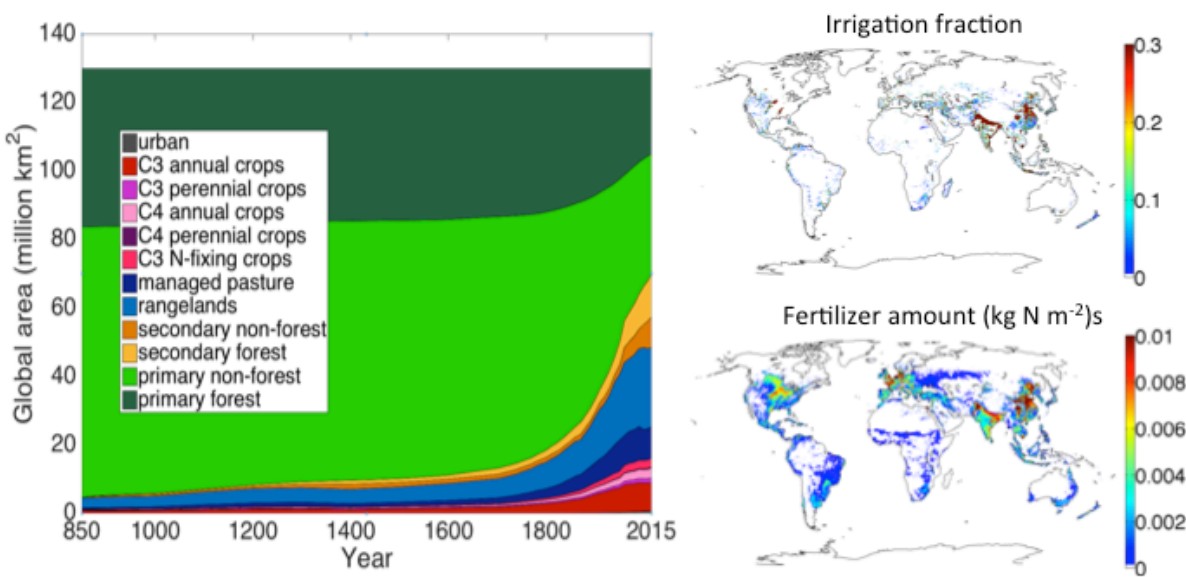

**Figure 1**. Time-series of global land area occupied by each LUH2 land-use state from 850 to 2015 (left). Note that extensions to 2100 for all of the ScenarioMIP SSPs will also be provided. Fraction of each 0.25$^{\circ}$ grid-cell that is irrigated in year 2015 (top right). Fertilizer applied in year 2015 (bottom right).

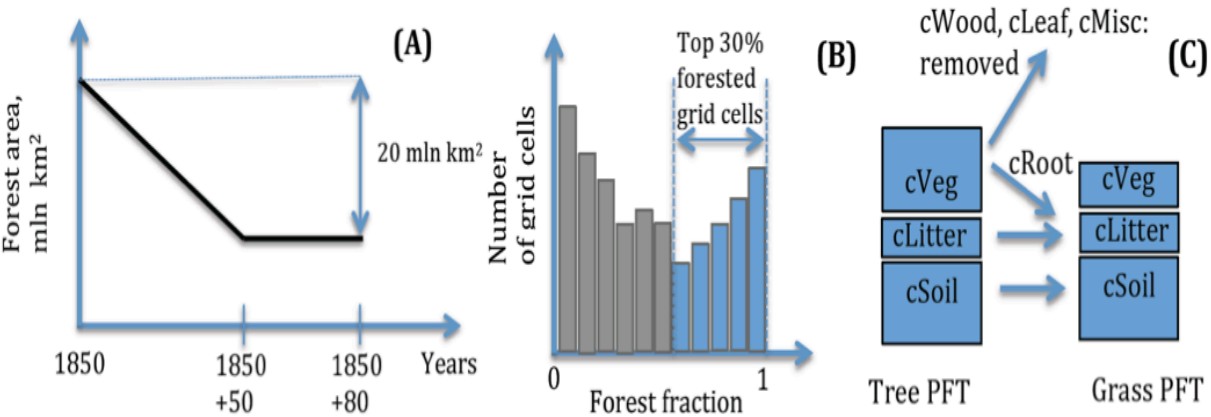

**Figure 2.** A schematic of experimental setup in the *deforest-glob* experiment. (A) Scenario of forced changes in the global forest area. (B) Sorting and selection of the grid cells that should be deforested. (C) Transition of carbon pools after deforestation.

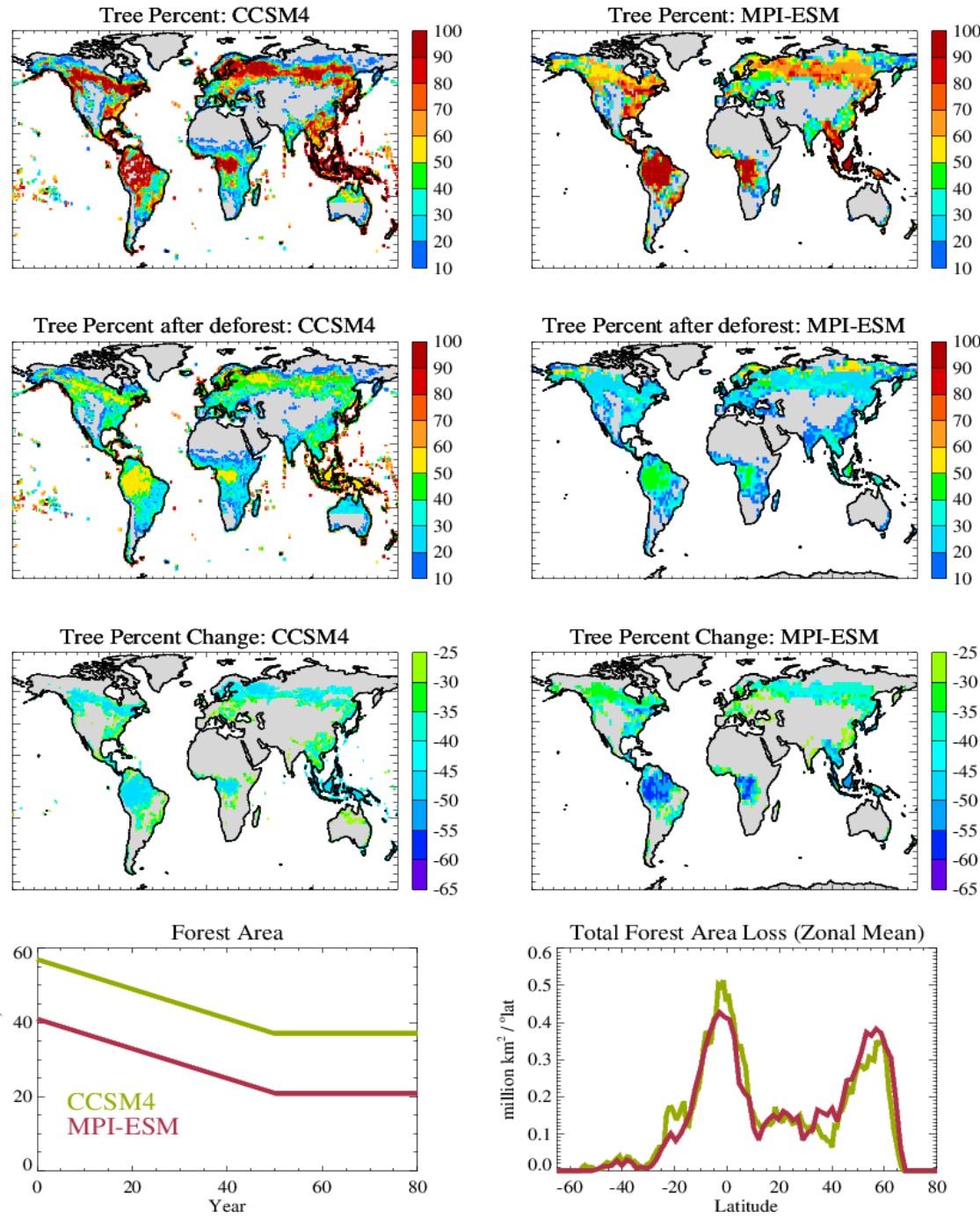

**Figure 3:** Sample maps of fraction of grid cell covered by trees at the start of the idealized deforestation simulation, after idealized deforestation (year 50), and the change in tree fraction by the end of the deforestation period. Time series of forest area and zonal mean forest area loss are also shown. Examples are shown for two typical CMIP5 models with strongly differing initial forest cover. Even with the different initial forest cover, the deforestation patterns and amounts are broadly equivalent across the two models.

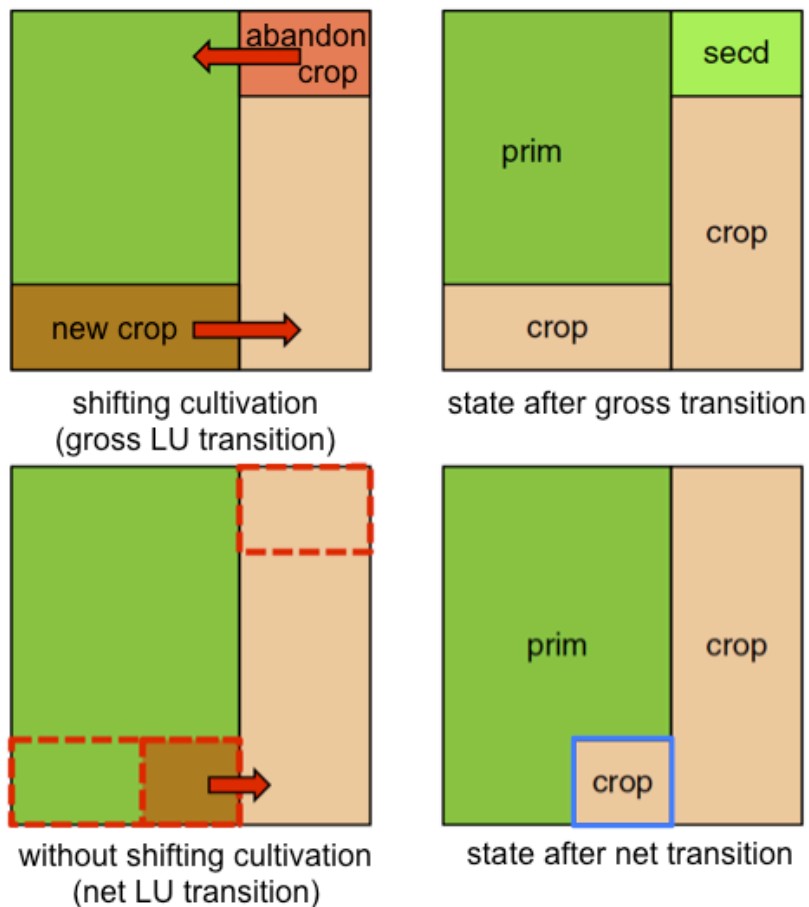

**Figure 4.** Schematic diagram showing difference between inclusion of shifting cultivation (gross transitions) versus exclusion of shifting cultivation (net transitions). Where shifting cultivation is included (upper row), new cropland (or pastureland) is taken (deforestation) from primary land ('prim') and abandoned to secondary land ('secd') in parallel within a grid cell. In this case carbon fluxes, for example, are captured for each transition. Where shifting cultivation is not represented (lower row), only the difference of new cropland minus abandoned cropland (represented by crop area outlined in blue in bottom right figure) undergoes a transition to cropland and no cropland is abandoned to form secondary land. In this case, a smaller grid cell area fraction is affected by LUC. Adapted from Figure 1 of Stocker et al. (2014).

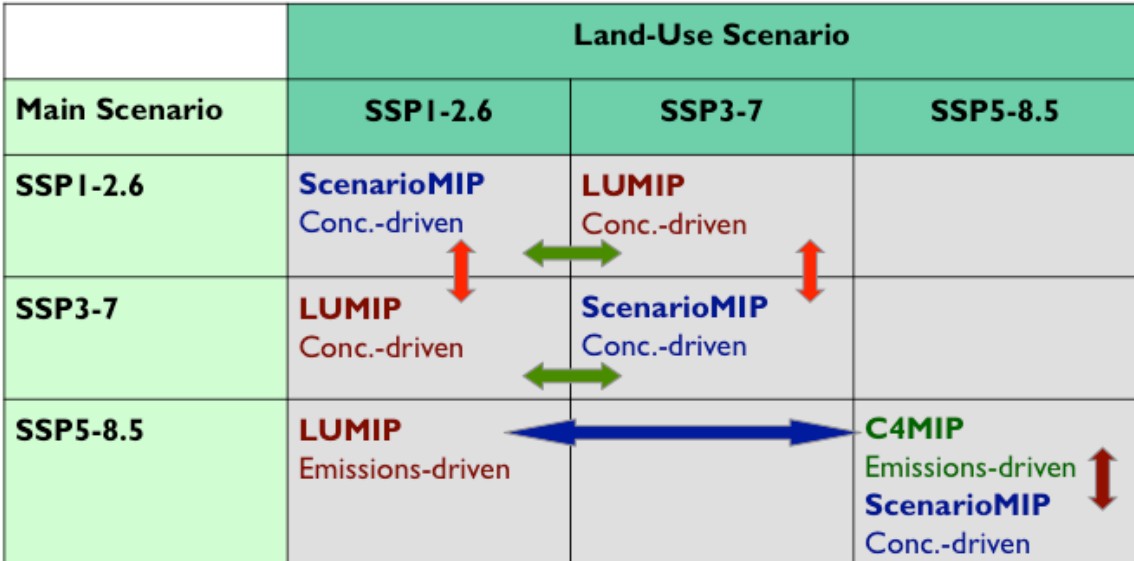

**Figure 5**: Schematic describing the future land-use policy sensitivity experiments.  Green arrows indicate set of experiments that permit analysis of the biogeophysical climate impacts of projected land use and enable assessment of land management as a regional climate mitigation tool.  Red arrows indicate set of experiments that allow study of how the impact of land-use change differs at different levels of climate change and at different levels of $CO_2$ concentration.   Blue arrow indicates set of experiments that will enable quantification of the full effects of a different land-use scenario through both biophysical and biogeochemical processes. Brown arrows indicate set of experiments that allow quantification of the effects of the climate-carbon cycle feedback on future $CO_2$ and climate change.

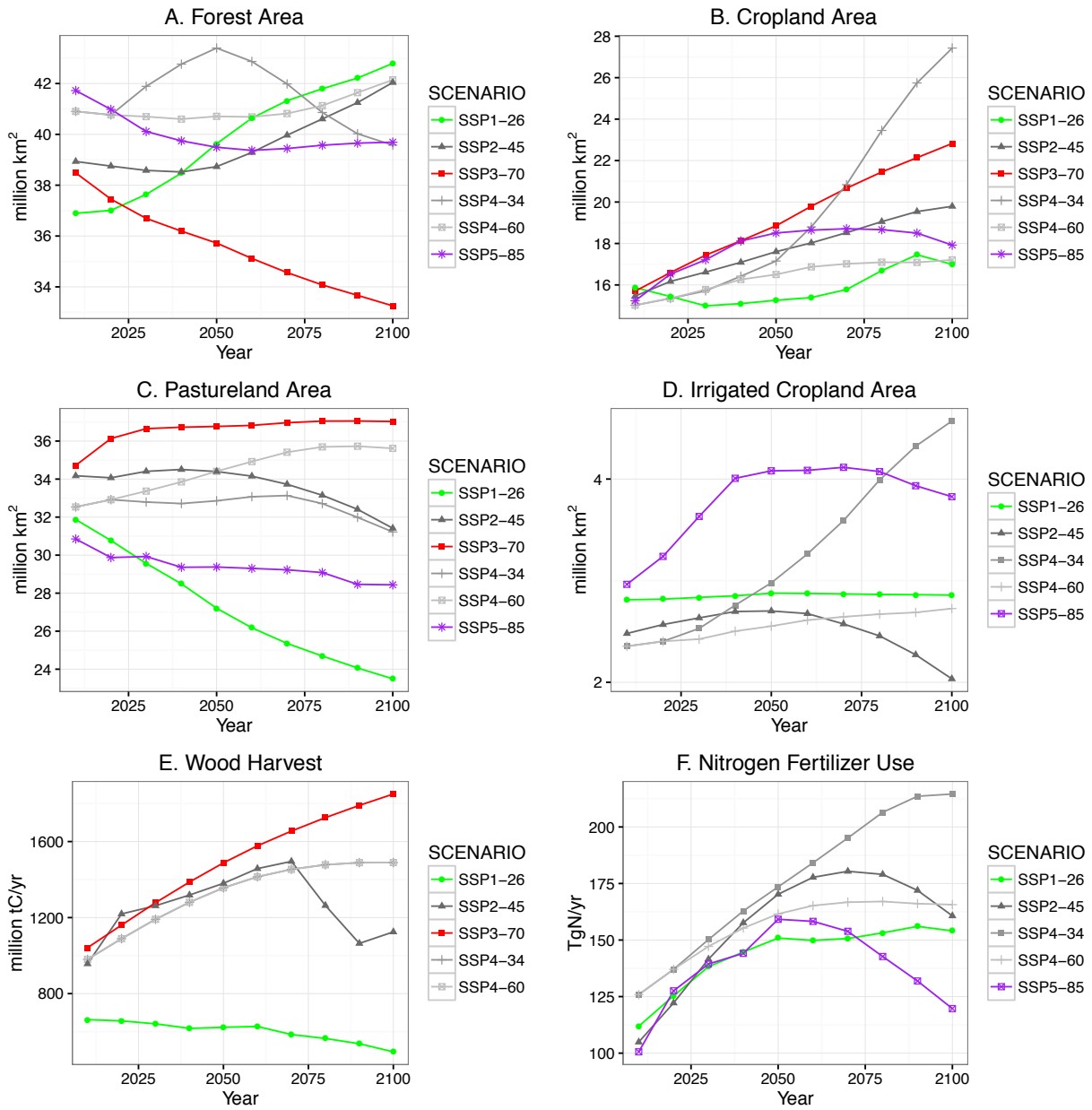

**Figure 6**: Global time series of land cover (A), land use (B, C, E), and land management (D, F) for the future simulations. Lines indicate SSP-RCP scenarios chosen for ScenarioMIP, with colored lines representing scenarios with specific LUMIP experiments. Data is provided by the IAM community (see Popp et al. 2016 for more details). Data will be harmonized to ensure consistency between the end of the historical period and the beginning of the projection period for each of the scenarios. Note that not all IAMs predict all the LUH2 land management quantities (e.g., wood harvest is missing for SSP5-8.5). The missing land management variables will be generated during the harmonization process in a manner that is consistent with the underlying scenario.

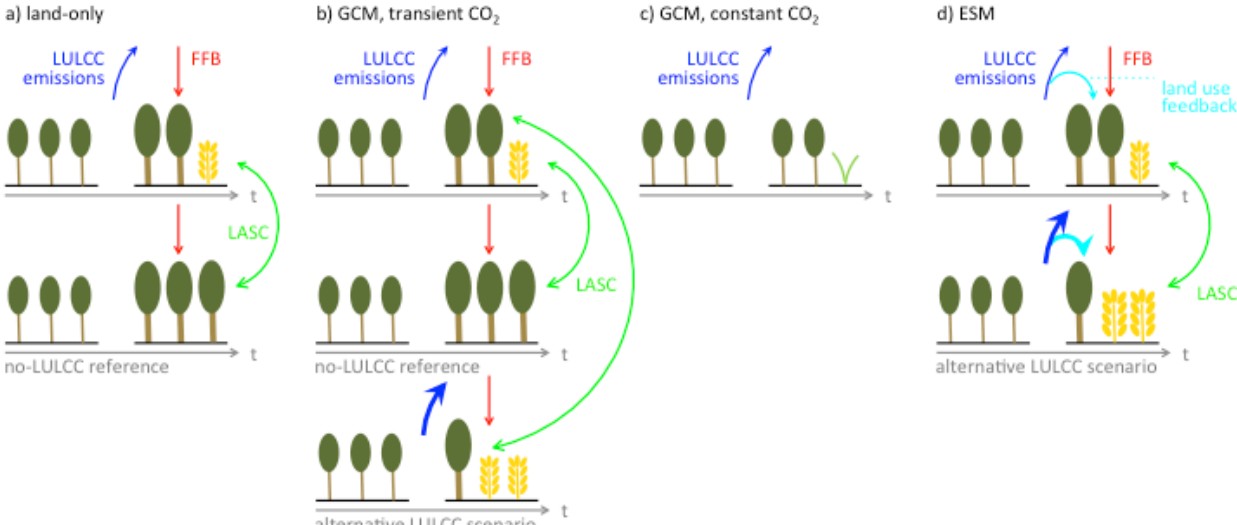

**Figure 7**: Illustration of the different setups used in the LUMIP experiments, using the example of forest replacement by cropland or grassland. The loss of additional sink capacity (LASC) is a factor when environmental conditions change transiently, which is the case when historical $CO_2$ concentrations, which implicitly include increases in $CO_2$ due to fossil-fuel burning (FFB) and LULCC, are prescribed from observations. Prognostic LULCC emissions are directly "seen" by the terrestrial vegetation (natural and anthropogenic) only in the ESM setup, where $CO_2$ is interactive. In this case, a fraction of the LULCC emissions is taken up again by the vegetation ("land-use carbon feedback"). Note that only atmospheric $CO_2$ is prescribed in a-c, while other environmental conditions feed back with LULCC's biogeophysical effects.

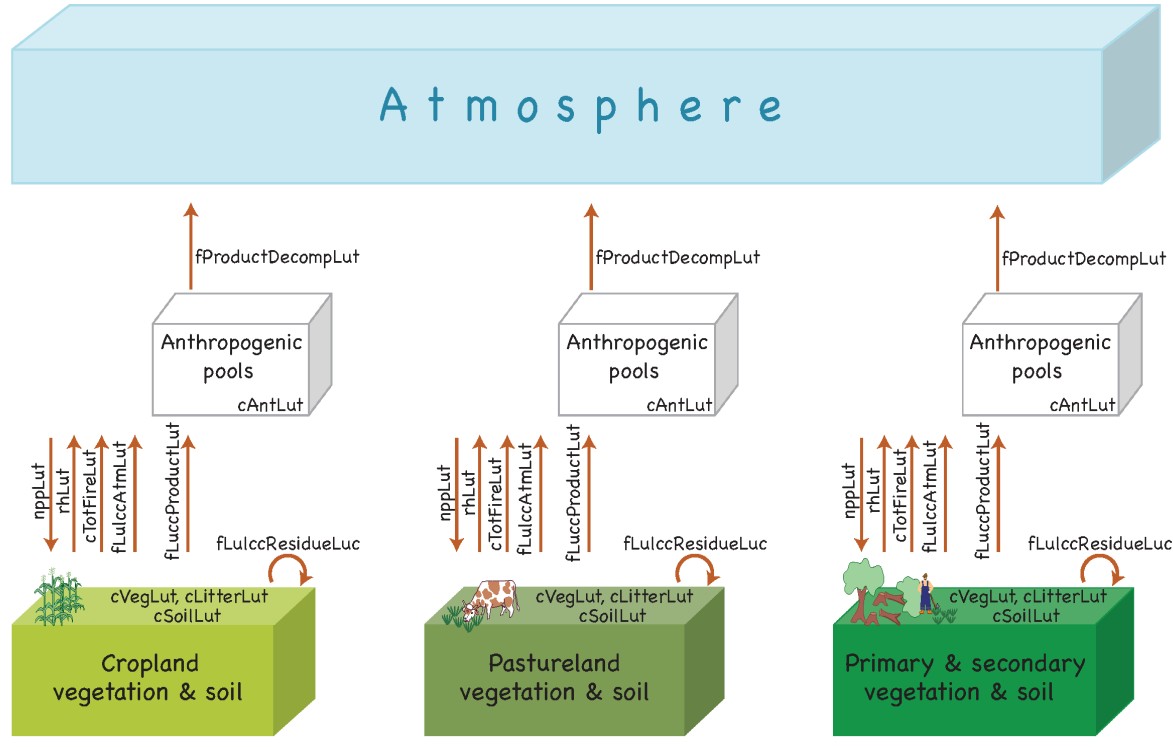

**Figure 8.** Exchanges and transfers affecting storage of biogeochemical constituents in land models under LULCC. Variable descriptions can be found in Table 5. Urban tile not shown, but if carbon fluxes are calculated on a particular model's urban tile, then these fluxes should be reported for urban tile as well.

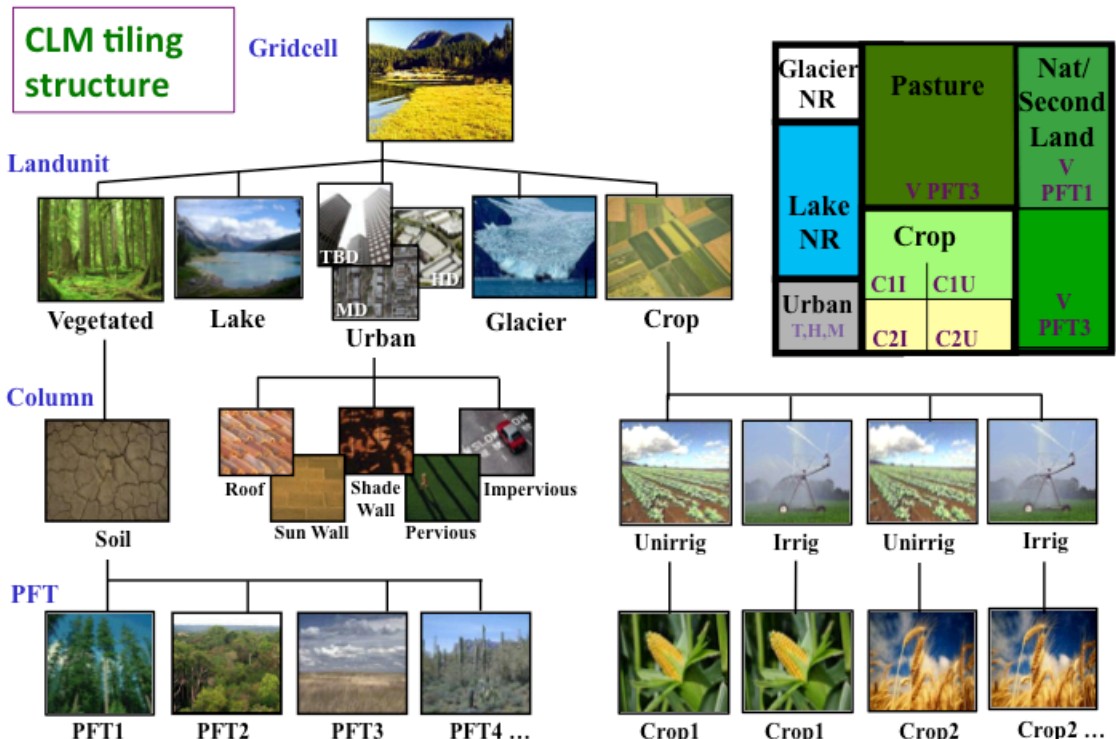

**Figure 9.** CLM tiling structure (Figure 8, Oleson et al. 2013). Subgrid aggregation: PSL = Vegetated land unit including all PFTs and bare soil; CRP = Crop land unit including all crop types irrigated (I) and non-irrigated (U); PST = not explicitly represented in CLM, report as missing value; URB = weighted average of Tall Building District, High Density, and Medium Density types in Urban landunit. Glacier and Lake are not reported.

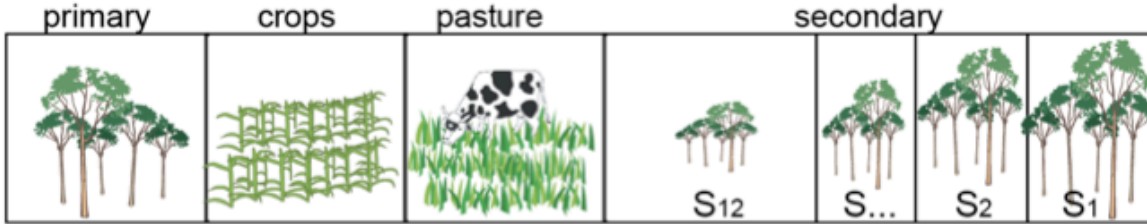

**Figure 10.** In the GFDL ESM2M and ESM2G CMIP5 simulations each grid cell has up to 15 land tiles, including lakes, glaciers, croplands, pasturelands, primary, and up to 10 secondary vegetation tiles. All GFDL models use gross transitions from the LULCC scenarios. The secondary vegetation tiles are generated by wood harvesting (primary to secondary and secondary to secondary transitions) as well as by agricultural abandonment (croplands to secondary and pastures to secondary transitions). Each land-use tile has its own C anthropogenic pool and separate above- and below-ground C stores. For LUMIP, all variables on primary and secondary tiles will be aggregated and reported under the PSL tile. Urban is not represented and will be reported as missing values. Glaciers and lakes are not reported.