# Peer review of "The Land Use Model Intercomparison Project (LUMIP) contribution to CMIP6: Rationale and experimental design"

_Geoscientific Model Development, 2016_

## Short Comment (SC1) · 13 Apr 2016

Dear authors,

In agreement with the CMIP6 panel members, the Executive editors of GMD would like to establish a common naming convention for the titles of the CMIP6 experiment description papers.

The title of CMIP6 papers should include both the acronym of the MIP, and CMIP6, so that it is clear this is a CMIP6-Endorsed MIP.

Additionally, we strongly recommend to add a version number to the MIP description. The reason for the version numbers is so that the MIP protocol can be updated later, normally in a second short paper outlining the changes. See, for example:

[Figure]

http://www.geosci-model-dev.net/special_issue11.html,

Good formats for the title include:

'XYZMIP (v1.0) contribution to CMIP6: Name of project'

or

'Name of Project (XYZMIP v1.0) contribution to CMIP6'

If you want to include a more descriptive title, the format could be along the lines of,

'XYZMIP (v1.0) contribution to CMIP6: Name of project - descriptive title'

or

'Name of Project (XYZMIP v1.0) contribution to CMIP6: descriptive title.'

When you revise your manuscript, please correct the title of your manuscript accordingly.

Yours,

Astrid Kerkweg

---

## Referee Comment (RC1) · A. Pitman (Referee) · 26 Apr 2016

Land use change is a necessary component of CMIP6 and a thorough and well argued case is made in this paper on how it should be done. Overall, this is a hugely ambitious MIP but one that if pulled off with a decent number of modelling groups would make profound strides forward. I think it will confront modelling groups and implementing these experiments will be challenging. But these experiments seem to be well thought through, appropriately designed and effectively described. My recommendation is therefore accept with minor revisions.

In the text from line 60 to 83 I got rather lost as to the argument. For example, the link between the sentences on line 69 seems opaque. The sentence starting "Levis" is about crop modelling, the previous sentence is about irrigation. I know what you are

trying to say but it does not really follow logically through this paragraph.

Line 111. Adaptation is local in most instances. I think its a big call to suggest CMIP6 models can inform us about adaptation given their spatial resolution. Maybe the best way to argue this is that LUMIP might provide approaches to this question at far higher resolution in RCMs?

Lines 112-119 are well stated. Hugely ambitious but well stated. The #5 does not really seem to fit to me however - although it is an important question. I am not proposing any changes but it might be worth a little more rationale?

Line 128 - I got confused here. I am not clear what the text "did not translate as such in land-dover data sets" really means

Line 130-135 - just a comment. This is the Porsche of LULCC science. I remain fearful that for most CMIP6 models the sophistication of the science presented here will disengage groups. A response "no it won't" is fine and time will tell.

Lines 219-225 0 I really did not know what you were trying to get across here.

Line 225 and 226 - I was confused here too. If the experiment is "constant land use" and you define fixed land use for a "relevant year" that implies to me you change land use annually and that implies anything but "fixed". Some clarification would be helpful.

Line 263-266 - this is a really important and valuable requirement.

Line 276, 282, 314 Figure X means what ?

Most of Section 2 is pitched at a good level of detail - balancing information that a reader might want with what a modeller doing the experiment might want. I do not think I could implement the experiments from this document - nor do I think that is a sensible thing to attempt. Is there going to be some place where full instructions will be given?

As someone who has worked in this place I can see the value of the different hierachy

of experiments - with coupled, uncoupled etc. I wonder if that should be explained for the non expert - why your experiments are constructed in the way they are. I know this would be clear to the authors but it might not be to a non-land cover modeller?

Line 474 - model evaluation is testing your model against observation. Model benchmarking is asking the question how well a model should perform given the information content in the forcing. I do not think they should be confused although I acknowledge they most certainly are in the community. You could resolve this by simply saying "need to improve diagnostics for land surface model evaluation and/or benchmarking in general".

line 495 - please no! Not student t-tests for LULCC. At the least you need a Findell test but there is far more to it and you need to account for field significance.

Line 555-557 - It is great to see coupling strength in here and a sensible solution implemented

Line 568-70 - seems vague. I appreciate you cannot resolve all aspects of this paper but this seemed particularly vague on extremes.

Line 600- is the reporting of subgrid variables a request or a requirement. I think it should ideally be required but that might put considerable stresses on many groups in terms of data handling. No specific recommendation here, but suggesting it should be clearer.

Minor edits

Line 3 First sentence of abstract does not make sense. Add "changes" after large

Line 19 "with respect to-" does not make sense.

Line 21 - The acronyms do not necessarily make sense to some readers and I think might be better avoided in the abstract. I do not know what ScenarioMIP is (!) and perhaps I should.

Line 59 - is this correct? 40% of the total radiative forcing? I would have guessed its 40% of the change in RF.

Line 66 "Other examines are numerous" is not a sentence.

―――――――――――――――

---

## Referee Comment (RC2) · P. Dirmeyer (Referee) · 1 May 2016

General comments:

Realizing this is primarily a "documentation" paper and not a "results" paper, my comments are mainly regarding clarity and completeness of description.

The NCAR Last Millennium Ensemble (LME) is not mentioned anywhere in this manuscript, but it is a natural antecedent to much of what is described here and it seems to me it would be handy to reference (e.g., ca. line 122; Otto-Bliesner et al. 2016).

Specific comments:

L3: "...large to..." - It appears one or more words are missing.

L19: "...respect to-." - Likewise, seems words are missing.

L41-43: Clarify: effect on global MEAN air temperature is small.

L98-102: Expand acronyms.

L200: Good to cite previous recent works regarding climate impacts of global deforestation (e.g., Davin and Noblet-Ducoudré 2010) and remote climate impacts of tropical deforestation (e.g., Snyder 2010, Badger and Dirmeyer 2016).

L228: Apparently more missing words, "...level if."

L263: I well understand and appreciate the issues of providing guidance to the execution of model runs in MIPs, but wouldn't it be good to declare an avenue for consultation - a wiki or something - to assist the groups "to make their own decisions..."?

L314: "Figure X" needs a number.

L321: Also cite Badger and Dirmeyer (2015) in this regard.

L409: Change "i.e.," to "e.g.,"

L476: Should cite the most recent effort at land model benchmarking - PLUMBER (Best et al. 2015).

L544-46: There have been investigations of the effect of land-atmosphere coupling on land use change responses. In particular, Kumar et al. (2013) developed a clever method to extract the land use change impact in CMIP5 simulations where multiple climate change factors were convolved in each RCP.

Sec 4.2: The existence variable output lists is mentioned then glossed over - please give a direct link to a list of variables (what is the "LUMIP CMIP6 variable request"?) or list them in supplemental tables in this paper. This is an important detail.

Figure 9 is not cited in text.

References:

Badger, A. M., and P. A. Dirmeyer, 2015: Climate response to Amazon forest replacement by heterogeneous crop cover. Hydrol. Earth Sys. Sci., 19, 4547-4557, doi: 10.5194/hess-19-4547-2015.

Badger, A. M., and P. A. Dirmeyer, 2016: Remote tropical and sub-tropical responses to Amazon deforestation. Climate Dyn. 46, 3057-3066, doi: 10.1007/s00382-015-2752-5.

Best, M. J., and co-authors, 2015: The plumbing of land surface models: benchmarking model performance. J. Hydrometeor., 16, 1425-1442, doi: 10.1175/JHM-D-14-0158.1.

Davin, E. L., and N. de Noblet-Ducoudré, 2010: Climatic impact of global-scale deforestation: Radiative versus nonradiative processes. J. Climate, 23, 97–112, doi: 10.1175/2009JCLI3102.1.

Kumar, S., P. A. Dirmeyer, V. Merwade, T. DelSole, J. M. Adams, and D. Niyogi, 2013: Land use/cover change impacts in CMIP5 climate simulations: A new methodology and 21st century challenges. J. Geophys. Res., 118, 6337–6353, doi: 10.1002/jgrd.50463.

Otto-Bliesner, B. L., E. C. Brady, J. Fasullo, A. Jahn, L. Landrum, S. Stevenson, N. Rosenbloom, A. Mai, and G. Strand, 2016: Climate variability and change since 850 C.E.: An ensemble approach with the Community Earth System Model (CESM), Bull. Amer. Meteor. Soc. (in press).

Snyder, P. K., 2010: The influence of tropical deforestation on the Northern Hemisphere climate by atmospheric teleconnections, Earth Interact., 14, 1–34, doi:10.1175/2010EI280.1.
* * *

---

## Referee Comment (RC3) · A. Di Vittorio (Referee) · 3 May 2016

Review of gmd-2016-76: The Land Use Model Intercomparison Project (LUMIP): Rationale and experimental design

The authors present the rationale and experimental protocol for the upcoming LUMIP. It appears that they intend this to be the primary reference document for the modeling groups participating in LUMIP. They first provide a comprehensive background and context with respect to CMIP6, and then describe an idealized experiment, a set of historical land-only experiments, and a set of future experiments. They then introduce a plan to develop metric, discuss challenges to analyses, and describe linkages to other MIPs that will enable studies relating LULCC to land-coupling strength and extremes. The paper concludes with a description and examples of subgrid data reporting for

LUMIP.

The introduction and context are thorough and compelling, and the set of experiments is quite comprehensive and ambitious. I am impressed by this expression of the tremendous effort put forth by the LUMIP team (and others who may have contributed).

I have reviewed this text as if I were a modeler expecting to participate in LUMIP. For the most part, the authors have done a good job explaining what is expected of the LUMIP participants. Apart from some clarification and additional detail, there is only one potentially major issue, and a few minor ones, that should be addressed prior to publication. I have summarized the main points here, with additional detail found in the specific suggestions and comments section below:

1) The experiments and the required simulations need to be presented in full, so that modeling teams can use this document for direct guidance. This means that the parent/control simulations shared with other MIPs need to be described with the relevant land configurations so that the experiments using LUMIP-specific simulations can be clearly presented or inferred. The tables 1-3 need to include all relevant sims, with the MIP-shared ones clearly marked, and the tier clearly marked. These tables will be an important reference. Please note that a simulation is not an experiment, and in most cases multiple simulations (at least on plus a control) are required to constitute an experiment. Of course, every little detail and contingency cannot be included here, and the authors do thankfully provide an online venue for further clarifications, updates, and details.

2) The potentially major issue involves the default land configuration for LUMIP and the rest of CMIP6. There are several aspects that should be addressed, but the one I am most concerned about is the use of gross (intra-annual) LULCC transitions as the default in models with this capacity. To make the most robust comparison across models, gross transitions should not be used, except as part of LUMIP to examine the consequences of including them. It is acknowledged that including gross transitions

can have a large impact on the carbon cycle, CMIP5 has already shown how including gross transitions can create carbon cycle outliers, the biogeophysical impacts of including gross transitions in models are not well established, and the gross transitions are probably the most uncertain component of the land use/cover data. Furthermore, I expect that still only a minority of models will be able to use the gross transitions. These are related notes that I don't expect to be dealt with here or in CMIP6, but should be thought about for future model comparisons: Irrigation and fertilization and other land management activities also raise red flags in this regard, but these may have smaller and more local effects than gross transitions, and so may be of less concern when comparing against models that do not have crop or management components. But gridded nitrogen application data are also very uncertain, as are nitrogen model components in general. Prognostic biogeography is another capacity that should be turned off for general model comparison (until most models have it, anyway), and turned on (in something like LUMIP) for examining the differences it generates (I understand that this would pose challenges for teams to do additional, separate model spin-ups for the two configs, but one may ask what the utility is of additional model comparisons in which the models continue to diverge in basic capacity and initial state).

3) The experiment to explore the effects of fertilization is not complete. This might have been a practical consideration, but it does not independently address both primary fertilization causes of differences in crop growth: area and application rate. questions regarding nitrogen application rates loom just as large as questions regarding fertilized area, and it would be useful to include the other two complementary sims: constant area with changing rate, and constant rate with changing area. Irrigation could be subjected to a similar set of sims, but it is less clear what it would mean to impose a constant irrigation rate because it varies year to year, usually based on environmental conditions and need.

4) What is the recommended protocol for using the new forest/non-forest area data in the non-idealiized sims and the rest of cmip6? You describe how different types of models (i.e. different initial forest area, different definitions of forest area, (non-)prognostic biogeography) should deal with forest in the idealized sim, but give no guidance on how different types of models should deal with the new forest area input. Just having the forest area input without guidance could still cause considerable divergence in land cover across models. I suggest presenting a recommended protocol for using the forest area data so as to minimize such divergence.

5) Please discuss the role of uncertainty in the LULCC data. There is a short section on uncertainty in atmospheric forcing data, and uncertainty in LULCC data is just as relevant, yet less understood. Addressing such uncertainty is beyond the scope of LUMIP, but the topic needs attention called to it because it will have to be addressed in the future.

specific suggestions and comments:

abstract

line 1: "...large [?changes?] to the..."

lines 12-13: not sure what you mean by "relative to fossil fuel emissions." see comment for line 677 below.

line 19: unfinished sentence "...with respect to ???"

line 20: How does this relate to the previous sentence? Are you only presenting activity (2)?

lines 18-21: I suggest expanding/explaining the acronyms here.

line 27: what is "a new subgrid land-use tile data request?" Does this mean you are also presenting activity (1)?

introduction

line 38: "...climate are relatively..."

line 68: "...expansion have likely..."

line 71: NEE is also a surface flux, albeit a net one - maybe use "seasonality of mass and energy surface fluxes" or something similar

line 80: "...climate has led to open..."

line 93-102: expand these acronyms on first time use

lumip activities

line 111: I suggest you separate out this third question, which takes additional work beyond what is required for question 2.

line 115: not sure what you mean by "relative to fossil fuel emissions." see comment for line 677 below.

line 124: It would be useful to define a protocol for using the forest/non-forest data in the non-idealized experiments. The protocol could be similar to that outlined for the idealized deforestation experiments, which acknowledges differences between prognostic and non-prognostic biogeography models and differences in initial forest cover among models. For example, the protocol could focus on matching annual changes in forest cover, with all the prognostic biogeography models including biogeographical changes for matching in the historical period, but not including them for matching in the future period (because the IAMs do not incorporate biogeographical effects on land cover in their scenarios). It would make sense for all the models to do the lumip sims without the prognostic biogeography (and then add sims to explore the effects of biogeography), but this would require additional sims to replicate those shared by other mips. this is something to think about for future mips.

lines 138-142: are the luh2 wood harvest data by volume/mass, or by area? or are both provided? can all the LUMIP participants deal with wood harvest mass? if both are provided, you may want to recommend (or request) that groups use the volume/mass data.

line 166: suggestion: ". . .variables on [individual] land-use tiles [within grid cells]. . .," or maybe 'distinct' or 'separate,' "multiple" is unclear

line 176: you may want to include a citation here as well, as evidence for this may not be widely acknowledged.

line 186: this example sounds more like land management (mowing vs not). maybe a better example of land use is whether forest is harvested or not (and rather than wood harvest be a management type, forest management options would include plantation vs tree selection vs clear cut). another land use example is whether grassland or shrubland is used for grazing/pasture or not, or whether cropland is annuals, perennials, orchard, or fallow (or whether there is cropland at all).

line 188: wood harvest is more like a land use, in that it describes the purpose (wood) for which humans exploit forest (or other land cover types). as mentioned above, there are several land management strategies that can be employed to achieve the land use of harvesting wood.

line 193: You may want to be clear that in this manuscript LULCC refers to ALCC only. I am not sure that this is generally the case, nor that it should generally be the case.

experimental design and description

lines 196-198: please revise and/or split this sentence to clarify it. also, expand DECK, as i think it is a first time use of the acronym

lines 198-199: awkward: ". . .coupled model idealized deforestation experiments. . ."

lines 201-202: ". . .the forced response of land-atmosphere fluxes to land cover change. . ."

lines 207-209: This information is incongruous and unclear. What does "request" mean? What do "tier 1" and "tier 2" mean? You also refer to tier 3 in table 3. What is tier 3?

line 210: what about section 2.1? lines 119-215 focus on the lumip experiments, and then you jump immediately, and unexpectedly, to a non-lumip discussion

lines 225-249: suggestion: separate paragraphs for general guidelines, 1850-specific guidelines, and >2100-specific guidelines also, rather than use "relevant year" and "constant land use year," pick a single, descriptive term to refer to the year that defines the "constant land use," such as "constant land use reference year," or something better

line 251: "...differences among CMIP6..."

line 255: need definition of "PI-control" - this can be done at first use, which may be line 221-220

phase 1 experiments

lines 268-270: it sounds like there is only one experiment also, table 1 includes only one simulation. it should also include the comparison/control simulation for the experiment, which appears to be the DECK picontrol.

line 276: is Fig X a supplemental figure? or should it be figure 2?

line 276: this should be included in table 1 also, it should be made clear that picontrol needs to be in equilibrium for several years prior to the branch, and how you intend to us picontrol as the control sim for the experiment. I am guessing that you intend to use an average of pre-branch picontrol years as the control for comparing the deforestation and post-deforestation years of the lumip sim (assuming that the picontrol isn't continuing in parallel, which would also work). however, 30 years of constant forest may not be enough time for the land carbon cycle to equilibrate, so comparison with pre-deforestation may not be robust. I suggest adding another 30 years to the post-deforestation part of the sim to ensure some stability for comparison with picontrol.

line 285: you may want to state "by the end of year t," which clearly includes models that change forest area throughout the year and models that make a single area change

during the year.

lines 288-296: It should be made clear that t=1850 is the initial state (i.e. t=0). especially in equation 2, where the t limits are not shown (maybe they should be).

lines 291 and 296: do you mean Ftot? and in line 296, this should be less than or equal to 20 M kmˆ2.

lines 292-293: shouldn't this be equation (2)? and it is currently duplicate.

lines 303-306: It should be requested that modeling teams report the annual spatial land type data (for diagnostics such as figure 2), and the global area of forest removed, so as to know which models were able to remove 20 M kmˆ2 of forest, and which ones were not able to do so.

line 305: "the examples shown in figure 2" should probably be in parentheses, as this phrase muddles the sentence a bit

line 332-335: land-hist is missing from the tier 1 list. it is still tier 1 even though this sim is also required for another cmip6 mip, which should be made more clear here and in table 2.

line 333: what is "X?" 13? and it looks like the period can be either 165 or 315

lines 334-335: the land-hist sims need to be described in detail, as it is the basis for all the other sims. for example, is the prognostic crop model part of this sim? are gross (intra-annual) lulcc transitions standard here?

lines 342-346: This is redundant, and as such, confusing. It sounds like something additional, but it isn't. It can be removed.

line 349: what do you mean by the "TRENDY" simulations? there is only one climate-related sim listed, and it is not indicated as a TRENDY simulation. Besides, these are LUMIP simulations, and it seems unnecessary to complicate things by calling one simulation a TRENDY simulation.

lines 350-351: not sure what "clean comparison" means. yes, the climate forcings will be the same, but there will still be land cover, land use, and land management differences among the models. And probably resolution differences as well. Different initial years and land states and how they came about will also introduce differences among the model outputs.

lines 372-377: this paragraph is evidence that the default for all cmip6 models should be either gross or net transitions. given that only some models can represent gross transitions, the high uncertainty of the gross transition data, and the accompanying uncertainty due to land cover translation (particularly non-forest), the default across all cmip6 should be net annual transitions. otherwise some models will have grossly different carbon estimates in all simulations and experiments. this means that the LUMIP simulation here should be land-grossTrans, where the gross transitions are enabled to explore their effects on surface mass and energy exchange.

lines 378-380: need to reference section 2.3.1 to tell reader that the appropriate GCM simulations will be available.

lines 381-389: Uncertainty in the driving land use/cover data poses the same challenge for comparison to observations. This needs to be acknowledged here as well, and I would expect it to be discussed more thoroughly in the LUH2 paper. Related to the uncertainty in the driving land use/cover data is the remaining uncertainty due to the translation of land use to land cover, which includes differences between land cover classes and plant functional types, the changes in non-forest land cover (which are not harmonized), and differences between the definition of forest in the LUH2 data and in the models, and how different models will implement the forest cover changes (e.g. prognostic vs. non-prognostic biogeography models). I don't think it is possible to explore the model sensitivity to land use/cover uncertainty in cmip6, but this exploration should be noted as a target for future cmips and land mips, with the potential for using additional land use/cover data sets to drive the models.

phase 2 experiments

line 395: "Historical" seems like an extra word here

line 399: describe all the relevant aspects of the cmip6 historical concentration-driven simulation. for example, what land use/management processes are included?

line 416: you may want to move your parenthetical note about ssp scenarios from line 420 to here. You should also include the relevant details of the parent sims here. e.g., what land use and land management activities are active.

line 422: land management isn't isolated in these experiments. the changes will be a combination of differences in land use, cover, and management (same issue in figure 3 caption). there may be individual pixels that can be extracted that have only land management/use/cover differences, but there will also be dependencies on the surrounding land what the total effects are for a given pixel. At the subgrid level this may work out for the crop data, but only if there are comparable crop areas between sims within the given pixels and only the management options are different (e.g. irrigation and fertilizer).

line 434: again, land management isn't isolated in these sims. and there will effects of surrounding land on a given pixel.

land use metrics and analysis plans

line 491: paired simulation analyses means that you need to ensure that your main control sims (which are shared with other mips) are well described in this paper as well the lumip specific ones, so that your lumip experiments are clear.

lines 527-531: redundant sentences

line 533: rfmip - another acronym needing expanding

lines 535-540: i suggest briefly describing the rfmip land experiment and how it complements lumip to make this paragraph more relevant.

lines 543-558: this is a good idea, and differences in land coupling strength among models may (or may not) also help identify where land use/cover/management may be different among models.

lines 592-594: This is a good idea, but I think that the forest and non-forest areas need to be separated out to replace the primary/secondary land category, to the extent possible (i expect that not all requested variables are kept track of at the forest/non-forest level, but some of them, such as carbon, are kept track of at the pft level in some models). for variables and models that do not distinguish between forest and non-forest, the primary/secondary value can be placed in one category with a flag in the other signifying that land cover is not segregated. This may not be practically feasible due to how the models store and write outputs, however, so it is something to consider for future comparisons, and maybe with more land cover types distinguished from each other.

lines 601-604: figures 6 and 7 don't seem to help much here, as they are not complete and clear about the variables (e.g., only biogeochemistry is shown, and fig 7 shows processes rather than variables). A table of all the requested variables, with the subgrid ones noted, would be more useful. please provide a link or a supplemental table of the full list of variables requested.

lines 651-654: please reference figure 9 if you want to include it.

line 674: for the future runs, land management isn't isolated (or will be extremely difficult to isolate, even at the subgrid level). you can get information about this from the historical land-only experiments, however.

line 677: not sure what you mean by "relative to fossil fuel emissions." It seems that the experiments are designed to quantify the effects of lulcc, in a more absolute sense, which can then be compared to the total emissions effects. I don't see quantification of fossil fuel effects only, nor outputs that would be lulcc effects relative to fossil fuel effects.

[Figure]

Tables and Figures

Table 1 please include other simulation required for the experiment it should be more clear that this is a tier 1 experiment

Table 2 it should be more clear which tier the experiments are in, and this should be noted in the same column for all. I suggest stating the tier at the beginning of each description or notes column, for each experiment. Or adding a narrow "tier" column on the right, with the appropriate number indicated. land-hist needs to be clearly marked as a sim that is shared by another mip. land-crop-nomanage: is all crop area constrained to 1850? so this is like a constant crop sim, and the pasture area and harvest can change over time? can irrigation amounts change? what about fertilization area? what is a "prognostic crop model" and how does it differ from what is used in the control sim? The description needs to be more complete as to what is different from the land-hist sim land-crop-nofert: i suggest two more sims to ask questions about the effects of changing area vs changing amounts: one with constant area and changing rate, and one with constant rate and changing area. land-netTrans: unclear what it means to maintain gross transitions in excess of net transitions also, the degree to which spatially gross transitions are included at coarser resolution depends on the upscaling process; the finer grid cells can be summed to get a net change for a coarser grid cell.

Table 3 the tier of each simulation needs to be clearly marked. i suggest adding rows for the control cmip6 sim, and the tier 2 and 3 ensemble members. tier 3 needs to be explained in the text. see comments above.

Table 4 This does not seem necessary, as this information, plus more, is directly available in the text.

Figure 3 Why note only 1 of the 3 additional lumip sims in the caption? note all or none. maybe state that the brown text are the additional sims.

Figure 4 I would classify wood harvest as land use, with various types of silvicul­ture/harvest (e.g. tree selection, clear cut, plantation, coppice) as land management. see the definitions you invoke in section 1.3.

Figure 5 This figure and its caption is not consistent with the section on net lulcc emis­sions. LULCC emissions are also "seen" by vegetation in prescribed transient CO2 sims also because the historical atmosphere data include all emissions (LULCC oc­curred historically) and the IAM projected CO2 emissions include their respective esti­mates of LULCC emissions. Furthermore, figure 5c also has LASC, even with constant CO2, because different land covers have different potential rates of carbon uptake.

Figure 9 not referenced by text

—————————————————

---

## Short Comment (SC2) · 21 May 2016

General Comments

For this review I used the CMIP Panel's letter to the MIPS as a guideline for my review. Below the bold parts are how I see LUMIP authors responding to the letter.

These contributions will detail: 1. the goal of the MIP and the major scientific gaps the MIP is addressing, and will specify what is new compared to CMIP5 and previous CMIP phases. "The Introduction covers this topic well, particularly section 1.1."

2. The contributions will include a description of the experimental design and scientific justification of each of the experiments for Tier 1 (and possibly beyond), and will link the experiments and analysis to the DECK and CMIP6 historical simulations. "Section

2 in the paper."

3. They will additionally include an analysis plan to fully justify the resources used to produce the various requested variables, and if the analysis plan is to compare model results to observations, the contribution will highlight possible model diagnostics and performance metrics specifying whether the comparison entails any particular requirement for the simulations or outputs (e.g. the use of observational simulators). "Section 3 in paper."

4. In addition, possible observations and reanalysis products for model evaluation are discussed and the MIPs are encouraged to help facilitate their use by contributing them to the obs4MIPs/ana4MIPs archives at the ESGF (see Section 3.3). "Discussed in section 2 and 3 of paper."

5. In some MIPs additional forcings beyond those used in the DECK and CMIP6 historical simulations are required, and these are described in the respective contribution as well. "Section 2 in the paper."

My summary is that the authors did a very nice job of responding to the CMIP Panel's directions. I have several specific comments below which I hope will further improve the paper.

Specific Comments

1. Page 2, Abstract, 1st line – Missing word. I think "changes" should go just after "large" and before "to the Earth surface"

2. Page 2, lines 29-30 – Should mention need for documentation of what the groups did to run the experiment. These details are at least as important to trying to follow the experimental design.

3.page 3, line 59 – 40% of radiative forcing – What is time period? When to when...

4. Page 6, line 142 – "industrial roundwood" – What is this? Please define.

5. Page 8, section 2 – I think there should be multiple mentions of the need for documentation of what was done and how by the modeling groups. Each group's land model is quite different from the others. The details will be very important if we are going to be able to figure out the results after the experiments are completed.

6. Page 10 – Several references to Figure X – line 276, 282, 299. Please insert correct figure number. 6B. Page 11, lines 314, 333

7. Page 13, top – This discussion is confusing to me. Cleanly discuss the various types of errors: model, forcing, observations.
* * *

---

## Author Comment (AC1) · 30 Jul 2016

**a.pitman@unsw.edu.au**

Land use change is a necessary component of CMIP6 and a thorough and well argued case is made in this paper on how it should be done. Overall, this is a hugely ambitious MIP but one that if pulled off with a decent number of modelling groups would make profound strides forward. I think it will confront modelling groups and implementing these experiments will be challenging. But these experiments seem to be well thought through, appropriately designed and effectively described. My recommendation is therefore accept with minor revisions.

**We thank Dr. Pitman for his positive comments about LUMIP in general. We acknowledge that the protocol is ambitious and that some of the simulations will be challenging to execute, but we are hopeful that through this paper and associated follow up organization/guidance that LUMIP will be a successful component of CMIP6.**

In the text from line 60 to 83 I got rather lost as to the argument. For example, the link between the sentences on line 69 seems opaque. The sentence starting "Levis" is about crop modelling, the previous sentence is about irrigation. I know what you are trying to say but it does not really follow logically through this paragraph.

**Rereading that paragraph, we see the point and have substantially rewritten it to improve its flow.**

Line 111. Adaptation is local in most instances. I think its a big call to suggest CMIP6 models can inform us about adaptation given their spatial resolution. Maybe the best way to argue this is that LUMIP might provide approaches to this question at far higher resolution in RCMs?

**Good point. We think removing the term adaptation from this question is more realistic. The input with respect to adaptation through LUMIP-related research would be indirect at best. We therefore removed the term adaptation here and in the abstract.**

Lines 112-119 are well stated. Hugely ambitious but well stated. The #5 does not really seem to fit to me however - although it is an important question. I am not proposing any changes but it might be worth a little more rationale?

**Yes, the questions that LUMIP intends to address are ambitious, but the intention is that LUMIP will draw on the breadth of expertise from a wide range of researchers who will utilize CMIP6/LUMIP simulation data. Regarding topic #5, we elect to retain this question as it is an area that has received some attention recently with respect to understanding the direct and indirect consequences of land-use change. We have reworded slightly to try to make this clearer: "the extent that the direct effects of higher CO2 concentrations on increases in global plant productivity are modulated by past and future land use.**

Line 128 - I got confused here. I am not clear what the text "did not translate as such in land-dover data sets" really means

**We have modified the sentence to try to make this clearer: "Note that land-cover data and forest/non-forest data, as well as land-use transitions, will be provided in the new dataset in order to help minimize misinterpretation of the land-use dataset that occurred in CMIP5 where, for example, the**

**strong afforestation in RCP4.5 was not captured in Community Earth System Model (CESM) simulations because of differing assumptions embedded within the CESM land use translator (a software package that translates the LUH data into CESM land-cover datasets) and the LUH dataset (Di Vittorio et al. 2014)."**

Line 130-135 - just a comment. This is the Porsche of LULCC science. I remain fearful that for most CMIP6 models the sophistication of the science presented here will disengage groups. A response "no it won't" is fine and time will tell.

**No it won't.  ☺  But seriously, we acknowledge again that LUMIP is ambitious, but we believe we have designed a set of experiments that will allow a range of researchers to make real progress in terms of our understanding of land cover and land management impacts on climate.  The LUH2 dataset is intentionally more comprehensive and contains more data layers than any single modeling group is likely to be able to ingest.  The goal is to help drive the whole field forward by pushing / encouraging groups to expand the scope of their models, possibly for CMIP6, but also beyond CMIP6.  The author list of the LUMIP paper, which consists of the LUMIP Scientific Steering Group, includes representatives of many of the major modeling centers that are working on land use issues.  These representatives have been heavily involved in both the experimental design and the production of the LUH2 dataset so none of this is going to come as a surprise to them.**

Lines 219-225 0 I really did not know what you were trying to get across here.

**We recognize that the definition of constant land use can be confusing.  We have extensively rewritten the entire section 2.1 to try to better clarify.**

Line 225 and 226 - I was confused here too. If the experiment is "constant land use" and you define fixed land use for a "relevant year" that implies to me you change land use annually and that implies anything but "fixed". Some clarification would be helpful.

**See above.**

Line 263-266 - this is a really important and valuable requirement.

**We agree.   This is especially critical for models that represent land-use history prior to 1850.**

Line 276, 282, 314 Figure X means what ?

**Apologies.  We have corrected the figure numbers throughout the text.**

Most of Section 2 is pitched at a good level of detail - balancing information that a reader might want with what a modeller doing the experiment might want. I do not think I could implement the experiments from this document - nor do I think that is a sensible thing to attempt. Is there going to be some place where full instructions will be given?

**The descriptions are meant to be as comprehensive as possible, but especially with the factorial set of simulations for and land-use and land-managements land-only experiments, the details will be somewhat specific to each modeling group.  We do/will maintain a website where more detailed instructions will be available where necessary and where we will maintain a forum for discussion of experimental setup.  This is already noted in the text:  "A forum for discussion of the experiments and for distribution of minor updates or clarifications to the experimental design will be hosted at the LUMIP website ([https://cmip.ucar.edu/lumip](https://cmip.ucar.edu/lumip))."**

As someone who has worked in this place I can see the value of the different hierachy of experiments - with coupled, uncoupled etc. I wonder if that should be explained for the non expert - why your experiments are constructed in the way they are. I know this would be clear to the authors but it might not be to a non-land cover modeller?

**The reviewer may have missed this, but we have already included a statement to this effect in the text. We believe that this provides sufficient justification for including both coupled and uncoupled simulations. "(a) The land-hist and land-noLu simulations will provide context for the global coupled CMIP6 historical simulations, enabling the disentanglement of the LULCC forcing (changes in water, energy and carbon fluxes due to land-use change) from the response (changes in climate variables like temperature and precipitation that are driven by LULCC-induced surface flux changes), though differences in the coupled model and observed climate forcing will need to be taken into account."**

Line 474 - model evaluation is testing your model against observation. Model benchmarking is asking the question how well a model should perform given the information content in the forcing. I do not think they should be confused although I acknowledge they most certainly are in the community. You could resolve this by simply saying "need to improve diagnostics for land surface model evaluation and/or benchmarking in general".

**We agree that the community needs to clarify the term benchmarking, but this is not the forum for that. We elect to remove the term benchmarking from this sentence completely.**

line 495 - please no! Not student t-tests for LULCC. At the least you need a Findell test but there is far more to it and you need to account for field significance.

**Good point. Mentioning t-tests was kind of a throwaway parenthetical that shouldn't have been included. We removed from the text and have included a new sentence mentioning the importance of field significance testing. "Lorenz et al. (2016) emphasize the importance of testing for field significance, especially in the context of evaluating the statistical significance of remote responses to LULCC."**

Line 555-557 - It is great to see coupling strength in here and a sensible solution implemented

**We agree. A more focused attention on the role of land-atmosphere coupling strength modulation of the land-use change signals is required. The opportunity to collaborate with LS3MIP on this will hopefully be productive.**

Line 568-70 - seems vague. I appreciate you cannot resolve all aspects of this paper but this seemed particularly vague on extremes.

**Not sure how to resolve this comment. Consideration of the impact of land use on extremes is a relatively new area of research and perhaps that explains the vagueness of the text. We felt it was important to highlight this as an area of analysis focus and believe that this paragraph serves that role. Took the opportunity to remove the term benchmarking, though.**

Line 600- is the reporting of subgrid variables a request or a requirement. I think it should ideally be required but that might put considerable stresses on many groups in terms of data handling. No specific recommendation here, but suggesting it should be clearer.

**We don't think that we can technically make anything a 'requirement' in terms of reporting, but the sub-grid request is a Tier 1 (highest priority) request in CMIP6.**

Minor edits

Line 3 First sentence of abstract does not make sense. Add "changes" after large

Line 19 "with respect to-" does not make sense.

Line 21 - The acronyms do not necessarily make sense to some readers and I think might be better avoided in the abstract. I do not know what ScenarioMIP is (!) and perhaps I should.

**Thanks for the edits, we have corrected them. For the final comment about not knowing what the other MIPs are, we elect to make the text clearer (that they are other CMIP6 MIPs) and retain mention of them in the abstract.**

Line 59 - is this correct? 40% of the total radiative forcing? I would have guessed its 40% of the change in RF.

**Correct, it is the change in RF. We have amended the sentence accordingly.**

Line 66 "Other examines are numerous" is not a sentence.

**We have corrected the sentence to: "Other examples of research indicating the importance of land management are numerous. "**
Realizing this is primarily a "documentation" paper and not a "results" paper, my comments are mainly regarding clarity and completeness of description.

The NCAR Last Millennium Ensemble (LME) is not mentioned anywhere in this manuscript, but it is a natural antecedent to much of what is described here and it seems to me it would be handy to reference (e.g., ca. line 122; Otto-Bliesner et al. 2016).

**We are familiar with the LME work and we looked through the LME paper again. It's not fully clear to us that the LME is directly relevant. Certainly, for the LME a historical land use reconstruction was generated, but it was kind of a mishmash of different datasets. For CMIP6, the LUH2 dataset will extend back to 850AD for the purpose of running last millennium simulations in PMIP. This is a positive development and will be discussed in the LUH2 document, but it seems tangential to what we are discussing in this paper. So, we have elected not to include mention of the LME here so as to avoid any confusion that LUMIP is really addressing the Last Millenium land use change topic.**

Specific comments:
L3: "...large to..." - It appears one or more words are missing.

**Corrected to "large changes to"**

L19: "...respect to-." - Likewise, seems words are missing.

**Corrected to with "respect to LULCC."**

L41-43: Clarify: effect on global MEAN air temperature is small.

**Corrected.**

L98-102: Expand acronyms.

**Done.**

L200: Good to cite previous recent works regarding climate impacts of global deforestation (e.g., Davin and Noblet-Ducoudré 2010) and remote climate impacts of tropical deforestation (e.g., Snyder 2010, Badger and Dirmeyer 2016).

**Later in the text, in Section 2.2.1 we cite several papers that have looked at global or regional deforestation. We add the suggested references there along with a phrase summarizing the results.**

L228: Apparently more missing words, "...level if."

**Corrected. Removed the word "if".**

L263: I well understand and appreciate the issues of providing guidance to the execution of model runs in MIPs, but wouldn't it be good to declare an avenue for consultation - a wiki or something - to assist the groups "to make their own decisions..."?

**Definitely, and this has always been the plan. We noted higher up in the paper that a forum will be available, but repeat that here, since initialization and defining constant land use for each model is likely to be among the more complex aspects of setting up the LUMIP simulations.**

L314: "Figure X" needs a number.

**Apologies. All figure captions have been corrected. Not sure how that error slipped through into the submitted version.**

L321: Also cite Badger and Dirmeyer (2015) in this regard.

**Done.**

L409: Change "i.e.," to "e.g.,"

**Done.**

L476: Should cite the most recent effort at land model benchmarking – PLUMBER (Best et al. 2015).

**Good point. Also added the Randerson et al. paper on carbon cycle metrics.**

L544-46: There have been investigations of the effect of land-atmosphere coupling on land use change responses. In particular, Kumar et al. (2013) developed a clever method to extract the land use change impact in CMIP5 simulations where multiple climate change factors were convolved in each RCP.

**We went back and reread the Kumar et al. (2013) paper and it doesn't seem to us that the role of land-atmosphere interactions is a primary focus of that study. We do cite the Kumar et al. (2013) paper earlier in the paper for it's argument that the LULCC impacts are uncertain. So, we elect not to add reference to the paper in the discussion of land-atmosphere interactions.**

Sec 4.2: The existence variable output lists is mentioned then glossed over – please give a direct link to a list of (what is the "LUMIP CMIP6 variable request"?) or list them in supplemental tables in this paper. This is an important detail.

**We have added a list of variables and noted that the list is subject to change. It's not clear to us at this stage how the CMIP6 variable request documents will be maintained/distributed so it is difficult to be clearer than that. However, there is a process that will/is being communicated to all modeling groups. "A list of requested land-use tile variables is shown in Table 5. However, this list is subject to change. Modelers should refer to the CMIP6 output request documents for the final variable list. "**

Figure 9 is not cited in text.

**Corrected.**
The authors present the rationale and experimental protocol for the upcoming LUMIP. It appears that they intend this to be the primary reference document for the modeling groups participating in LUMIP. They first provide a comprehensive background and context with respect to CMIP6, and then describe an idealized experiment, a set of historical land-only experiments, and a set of future experiments. They then introduce a plan to develop metric, discuss challenges to analyses, and describe linkages to other MIPs that will enable studies relating LULCC to land-coupling strength and extremes. The paper concludes with a description and examples of subgrid data reporting for LUMIP.

The introduction and context are thorough and compelling, and the set of experiments is quite comprehensive and ambitious. I am impressed by this expression of the tremendous effort put forth by the LUMIP team (and others who may have contributed). I have reviewed this text as if I were a modeler expecting to participate in LUMIP. For the most part, the authors have done a good job explaining what is expected of the LUMIP participants. Apart from some clarification and additional detail, there is only one potentially major issue, and a few minor ones, that should be addressed prior to publication. I have summarized the main points here, with additional detail found in the specific suggestions and comments section below:

1) The experiments and the required simulations need to be presented in full, so that modeling teams can use this document for direct guidance. This means that the parent/ control simulations shared with other MIPs need to be described with the relevant land configurations so that the experiments using LUMIP-specific simulations can be clearly presented or inferred. The tables 1-3 need to include all relevant sims, with the MIP-shared ones clearly marked, and the tier clearly marked. These tables will be an important reference. Please note that a simulation is not an experiment, and in most cases multiple simulations (at least on plus a control) are required to constitute an experiment. Of course, every little detail and contingency cannot be included here, and the authors do thankfully provide an online venue for further clarifications, updates, and details.

**In the Tables, we already reference the 'parent/control' simulations including names and the relevant MIP. Though LUMIP has been in consultation with these other MIPs during their experimental design,**

**LUMIP does not control how those simulations are setup or executed.  In our opinion, the tables are clear as is and adding the names of additional experiments that are not in the LUMIP request is more likely to confuse rather than clarify.  All modeling centers that participate in CMIP6 should be able to clearly reference these tables in combination with the other MIP documentation papers to execute the required simulations.   We did add information on the Tier of the parent / control simulation.  In all cases, LUMIP intentionally built off of Tier 1 simulations from other MIPs.**

2) The potentially major issue involves the default land configuration for LUMIP and the rest of CMIP6. There are several aspects that should be addressed, but the one I am most concerned about is the use of gross (intra-annual) LULCC transitions as the default in models with this capacity. To make the most robust comparison across models, gross transitions should not be used, except as part of LUMIP to examine the consequences of including them. It is acknowledged that including gross transitions can have a large impact on the carbon cycle, CMIP5 has already shown how including gross transitions can create carbon cycle outliers, the biogeophysical impacts of including gross transitions in models are not well established, and the gross transitions are probably the most uncertain component of the land use/cover data. Furthermore, I expect that still only a minority of models will be able to use the gross transitions.

**We acknowledge that the impact of gross transitions is uncertain, but we would also argue that including gross transitions in some fashion is more defensible that leaving it out completely.  In any case, LUMIP / CMIP6 cannot and does not control decisions on model configurations.  Those decisions are left to the modeling centers.  In the case of LUMIP, we hope that the additional data layers in the LUH2 dataset will spur groups to include additional relevant processes, but that remains up to each modeling center as they balance many competing research needs and foci.  That said, we have explicitly included a simulation in the set of land-only experiments (land-netTrans) that will allow LUMIP to assess the impact of gross versus net transitions in a multi-model context.**

These are related notes that I don't expect to be dealt with here or in CMIP6, but should be thought about for future model comparisons: Irrigation and fertilization and other land management activities also raise red flags in this regard, but these may have smaller and more local effects than gross transitions, and so may be of less concern when comparing against models that do not have crop or management components. But gridded nitrogen application data are also very uncertain, as are nitrogen model components in general. Prognostic biogeography is another capacity that should be turned off for general model comparison (until most models have it, anyway), and turned on (in something like LUMIP) for examining the differences it generates (I understand that this would pose challenges for teams to do additional, separate model spin-ups for the two configs, but one may ask what the utility is of additional model comparisons in which the models continue to diverge in basic capacity and initial state).

**Our response to this is similar to that stated above.  LUMIP cannot specify the configuration that each modeling center elects to utilize in CMIP simulations.  The potentially considerable differences in configuration across models will certainly complicate analysis, but there is no practical solution to this problem.  We agree that specific questions about specific aspects of land use on climate will require more targeted MIP efforts.  Clever use of the data can still be informative.  A main intent of the realistic historical or projection experiments is not to understand model differences but to understand the potential impact of land-use on current and future climate.  In this context, the structural differences across models is simply an element of uncertainty that cannot be reduced at this point.   Other simulations within LUMIP will be more useful in terms of assessing models relative to each other.  In particular, the idealized deforestation experiment is designed to allow direct comparison across models of the impact of deforestation.  Furthermore, the land-only land use experiments are designed for a multi-model assessment/comparison of the impact on surface fluxes of various aspects of land use over the historical period.**

3) The experiment to explore the effects of fertilization is not complete. This might have been a practical consideration, but it does not independently address both primary fertilization causes of differences in crop growth: area and application rate. Questions regarding nitrogen application rates loom just as large as questions regarding fertilized area, and it would be useful to include the other two complementary sims: constant area with changing rate, and constant rate with changing area. Irrigation could be subjected to a similar set of sims, but it is less clear what it would mean to impose a constant irrigation rate because it varies year to year, usually based on environmental conditions and need.

**Almost every one of the simulations listed in Table 2 could be expanded into several different experiments to tease out the relative contribution of specific aspects of land use/management.  The list that we have included is already very long and is likely to tax groups considerably to get them all done. Others have proposed additional simulations and at this point, our plan is not to expand this any further, but as the project moves forward over the next several years, we reserve the right to add additional optional experiments, which we will request/advertise through the LUMIP website and mailing lists.**

4) What is the recommended protocol for using the new forest/non-forest area data in the non-idealiized sims and the rest of cmip6? You describe how different types of models (i.e. different initial forest area, different definitions of forest area, (non-)prognostic biogeography) should deal with forest in the idealized sim, but give no guidance on how different types of models should deal with the new forest area input. Just having the forest area input without guidance could still cause considerable divergence in land cover across models. I suggest presenting a recommended protocol for using the forest area data so as to minimize such divergence.

**There will be another paper on the land use datasets themselves where guidance will be given. Preliminary guidance is also available through the LUMIP website.  We now note both avenues for guidance in the text.**

5) Please discuss the role of uncertainty in the LULCC data. There is a short section on uncertainty in atmospheric forcing data, and uncertainty in LULCC data is just as relevant, yet less understood. Addressing such uncertainty is beyond the scope of LUMIP, but the topic needs attention called to it because it will have to be addressed in the future.

**After discussing further with the LUMIP team, we have elected to add 2 additional simulations in the land-only set of simulations to provide a preliminary sensitivity analysis of uncertainty in historical land-use. We added the following text as well as ammendments to Table 2.**
**"To help address the issue of sensitivity to uncertainty in historical land-use forcing, two alternative historical land-use reconstructions have also been developed. These alternatives are based on same data sources, use same algorithms,  and are provided in same format as the reference LUH2 product, but  span range of uncertainty in the key historical input datasets for agriculture and wood harvest. Specifically, the 'high' reconstruction, assumes high historical estimates for crop and pasture and wood harvest, and the 'low' reference assumes low estimates for each of these terms, relative to the reference.  "**

specific suggestions and comments:

abstract

line 1: ". . .large [?changes?] to the. . ."

**Corrected.**

lines 12-13: not sure what you mean by "relative to fossil fuel emissions." see comment for line 677 below.

**See answer below.**

line 19: unfinished sentence ". . .with respect to ???"

**Corrected.**

line 20: How does this relate to the previous sentence? Are you only presenting activity (2)?

**We modified the text to : "In this manuscript, we describe the LUMIP activity (2), i.e., the LUMIP simulations …"**

lines 18-21: I suggest expanding/explaining the acronyms here.

**Unless we get guidance from GMD that this is required, we prefer the succinctness as stated in the abstract.**

line 27: what is "a new subgrid land-use tile data request?" Does this mean you are also presenting activity (1)?

**This is explained in detail in the body of the text. We amend the sentence slightly to increase clarity "describes a new subgrid land-use tile data request for selected variables (reporting data separately for primary and secondary land, crops, pasture, urban).**

introduction
line 38: ". . .climate are relatively. . ."

**Corrected.**

line 68: ". . .expansion have likely. . ."

**Corrected.**

line 71: NEE is also a surface flux, albeit a net one - maybe use "seasonality of mass and energy surface fluxes" or something similar

**Corrected.**

line 80: ". . .climate has led to open. . ."

**Corrected.**

line 93-102: expand these acronyms on first time use

**Done.**

lumip activities

line 111: I suggest you separate out this third question, which takes additional work beyond what is required for question 2.

**Done.**

line 115: not sure what you mean by "relative to fossil fuel emissions." see comment for line 677 below.

**See answer below.**

line 124: It would be useful to define a protocol for using the forest/non-forest data in the non-idealized experiments. The protocol could be similar to that outlined for the idealized deforestation experiments, which acknowledges differences between prognostic and non-prognostic biogeography models and differences in initial forest cover among models. For example, the protocol could focus on matching annual changes in forest cover, with all the prognostic biogeography models including biogeographical changes for matching in the historical period, but not including them for matching in the future period (because the IAMs do not incorporate biogeographical effects on land cover in their scenarios). It would make sense for all the models to do the lumip sims without the prognostic biogeography (and then add sims to explore the effects of biogeography), but this would require additional sims to replicate those shared by other mips. this is something to think about for future mips.

**The protocol for use of the forest/non-forest data will be provided in the LUH2 paper and associated documentation on the LUMIP website. We agree that it would be good to try to isolate the impact of dynamic biogeography, but we deem this outside the scope of the current LUMIP effort, which is already very extensive.**

lines 138-142: are the luh2 wood harvest data by volume/mass, or by area? or are both provided? can all the LUMIP participants deal with wood harvest mass? if both are provided, you may want to recommend (or request) that groups use the volume/mass data.

**Both are provided, but this is a topic for the LUH2 paper.**

line 166: suggestion: ". . .variables on [individual] land-use tiles [within grid cells]. . .," or maybe 'distinct' or 'separate,' "multiple" is unclear

**Text amended to "or selected key variables on separate land-use tiles within each grid cell (primary …"**

line 176: you may want to include a citation here as well, as evidence for this may not be widely acknowledged.

**Not sure what the reviewer wants here. In this section, we are only noting what WCRP Grand Challenges that LUMIP will be able to contribute to. This isn't the place to include references to work that shows whether or not LULCC has an impact in these areas. Those references are included elsewhere. The Trenberth and Asrar reference actually just refers to the document where that Grand Challenge is introduced. It probably makes more sense to remove it, so as not to be confusing.**

line 186: this example sounds more like land management (mowing vs not). Maybe a better example of land use is whether forest is harvested or not (and rather than wood harvest be a management type, forest management options would include plantation vs tree selection vs clear cut). another land use example is whether grassland or shrubland is used for grazing/pasture or not, or whether cropland is annuals, perennials, orchard, or fallow (or whether there is cropland at all).
line 188: wood harvest is more like a land use, in that it describes the purpose (wood) for which humans exploit forest (or other land cover types). as mentioned above, there are several land management strategies that can be employed to achieve the land use of harvesting wood.

**These are good points. There are grey areas, but the definition can certainly be better stated:**

**"Land cover refers to "the attributes of the Earth's land surface and immediate subsurface, including biota, soil, topography, surface and groundwater, and human (mainly built-up) structures", and is represented in land models by categories like forest, grassland, cropland or urban areas. Land use is the "purpose for which humans exploit the land cover"; e.g., a grassland may be left in its natural state, mowed, or utiilzed as rangeland for livestock. Land management refers to ways in which humans treat vegetation, soil, and water, and is captured in land models by processes such as irrigation, use of fertilizers and pesticides, crop species selection, or methods of wood harvesting (selective logging versus clear cutting)."**

line 193: You may want to be clear that in this manuscript LULCC refers to ALCC only. I am not sure that this is generally the case, nor that it should generally be the case. experimental design and description

**Good point. We have modified sentence to make this more clear.**

lines 196-198: please revise and/or split this sentence to clarify it. also, expand DECK, as i think it is a first time use of the acronym

**Done.**

lines 198-199: awkward: ". . .coupled model idealized deforestation experiments. . ."

**Rewritten to: "Phase one features a coupled model simulation with an idealized deforestation scenario that is designed to advance process-level understanding and to quantify model sensitivity to land-cover change impacts on climate and biogeochemical stocks and fluxes."**

lines 201-202: ". . .the forced response of land-atmosphere fluxes to land cover change. . ."

**Done.**

lines 207-209: This information is incongruous and unclear. What does "request" mean? What do "tier 1" and "tier 2" mean? You also refer to tier 3 in table 3. What is tier 3?

**We have modified the text to try to make this more clear. The Tier 3 experiment was a typo and has been removed.**
**"Details of the model experiments are described below. The full set of LUMIP experiments include:**
**•       Tier 1 (high priority): 500 years GCM/ESM; 650 years land-only**
**•       Tier 2 (medium priority): 500 years GCM/ESM; up to 1500 years land-only**
**Note that these totals only represent the LUMIP-sponsored simulations. LUMIP analysis requires control simulations from other MIPs, e.g., a pre-industrial control DECK simulation or a CMIP6 historical simulation. We note the required 'parent' simulation and responsible MIP, where applicable."**

line 210: what about section 2.1? lines 119-215 focus on the lumip experiments, and then you jump immediately, and unexpectedly, to a non-lumip discussion

**We changed the text to make this clearer.**
**"In this section, we begin with a discussion and recommendations on the specification of land use in CMIP6 Diagnostic, Evaluation and Characterization of Klima (DECK) and historical experiments and other MIP experiments (Section 2.1). Also in this section, we outline the full set of requested LUMIP experiments (Sections 2.2 and 2.3). LUMIP includes a two phase, tiered, model experiment plan."**

lines 225-249: suggestion: separate paragraphs for general guidelines, 1850-specific guidelines, and >2100-specific guidelines also, rather than use "relevant year" and "constant land use year," pick a single,

descriptive term to refer to the year that defines the "constant land use," such as "constant land use reference year," or something better

**We couldn't find a clean way to separate the paragraphs, but we did take the suggestion of referring to the constant land use year as the constant land use reference year.**

line 251: ". . .differences among CMIP6. . ."

**Corrected.**

line 255: need definition of "PI-control" - this can be done at first use, which may be line 221-220

**We think that the phrase already included in the text explains the pre-industrial control simulation sufficiently. Details, as with all the parent simulations, should be obtained within the CMIP6 paper by Eyring et al. (2016).**

phase 1 experiments

lines 268-270: it sounds like there is only one experiment also, table 1 includes only one simulation. it should also include the comparison/control simulation for the experiment, which appears to be the DECK picontrol.

**That is correct. We had been considering some regional deforestation experiments to go along with this experiment, but decided in the end not to include them. We have corrected the text to make it clear that there is only one idealized deforestation simulation. In Table 1, we add text to make it clear that the idealized deforestation simulation should be setup identically to the piControl simulation.**

line 276: is Fig X a supplemental figure? or should it be figure 2?

**Apologies, we mistakenly didn't get the numbering right for this figure. It has been corrected.**

line 276: this should be included in table 1 also, it should be made clear that picontrol needs to be in equilibrium for several years prior to the branch, and how you intend to us picontrol as the control sim for the experiment. I am guessing that you intend to use an average of pre-branch picontrol years as the control for comparing the deforestation and post-deforestation years of the lumip sim (assuming that the picontrol isn't continuing in parallel, which would also work). however, 30 years of constant forest may not be enough time for the land carbon cycle to equilibrate, so comparison with pre-deforestation may not be robust. I suggest adding another 30 years to the post-deforestation part of the sim to ensure some stability for comparison with picontrol.

**We have now noted in text that the run should be branched from at least 80 years prior to the end of the piControl and that is should be branched from a point of stability. We have not added 30 years to the end of the simulation, in the interest of keeping the number of simulated years as low as possible. We acknowledge that the carbon stocks will not necessarily be in equilibrium, but full equilibrium is not required in this case where we will mainly be evaluating relative changes across models.**

line 285: you may want to state "by the end of year t," which clearly includes models that change forest area throughout the year and models that make a single area change during the year.

**Done.**

lines 288-296: It should be made clear that t=1850 is the initial state (i.e. t=0). Especially in equation 2, where the t limits are not shown (maybe they should be).

**Corrected as suggested.**

lines 291 and 296: do you mean Ftot? and in line 296, this should be less than or equal to 20 M kmˆ2.

**Yes. We have corrected.**

lines 292-293: shouldn't this be equation (2)? and it is currently duplicate.

**Yes. Corrected.**

lines 303-306: It should be requested that modeling teams report the annual spatial land type data (for diagnostics such as figure 2), and the global area of forest removed, so as to know which models were able to remove 20 M kmˆ2 of forest, and which ones were not able to do so.

**We have amended the text to reflect this. Annual data on forest fraction is in the data request.**

line 305: "the examples shown in figure 2" should probably be in parentheses, as this phrase muddles the sentence a bit

**Done**

line 332-335: land-hist is missing from the tier 1 list. it is still tier 1 even though this sim is also required for another cmip6 mip, which should be made more clear here and in table 2.

**We have attempted to make this clearer and included mention that the land-hist simulation is required, even if LS3MIP is not completed by a particular modeling center.**

line 333: what is "X?" 13? and it looks like the period can be either 165 or 315

**Yes, it is 13, now included I text.**

lines 334-335: the land-hist sims need to be described in detail, as it is the basis for all the other sims. for example, is the prognostic crop model part of this sim? are gross (intra-annual) lulcc transitions standard here?

**We add some text to make it clear that the land-hist simulation should have the same land configuration as in the coupled CMIP6 historical simulation. We also note that additional land-hist configurations are possible if a group has more advanced land model and that groups can utilize that configuration additionally.**

lines 342-346: This is redundant, and as such, confusing. It sounds like something additional, but it isn't. It can be removed.

**We don't think that this text is redundant. This text covers the situation where a modeling group may wish to utilize their more advanced land model version, in addition to their coupled model version, for the full set of factorial experiments.**

line 349: what do you mean by the "TRENDY" simulations? there is only one climate related sim listed, and it is not indicated as a TRENDY simulation. Besides, these are LUMIP simulations, and it seems unnecessary to complicate things by calling one simulation a TRENDY simulation.

**Agreed. We have removed reference to TRENDY here.**

lines 350-351: not sure what "clean comparison" means. yes, the climate forcings will be the same, but there will still be land cover, land use, and land management differences among the models. And probably resolution differences as well. Different initial years and land states and how they came about will also introduce differences among the model outputs.

**We take your point, they are relatively clean, but there will be the usual challenges.  We remove the sentence.**

lines 372-377: this paragraph is evidence that the default for all cmip6 models should be either gross or net transitions. given that only some models can represent gross transitions, the high uncertainty of the gross transition data, and the accompanying uncertainty due to land cover translation (particularly non-forest), the default across all cmip6 should be net annual transitions. otherwise some models will have grossly different carbon estimates in all simulations and experiments. this means that the LUMIP simulation here should be land-grossTrans, where the gross transitions are enabled to explore their effects on surface mass and energy exchange.

**As noted above, we cannot dictate within CMIP6 what the default configuration for each model is.  We include the experiments to look at gross versus net in the land-only simulations to help us identify what impact the assumptions about whether or not to include gross (and how it is included) has.  You could probably identify problems with all sorts of assumptions that go into the treatment of land use in these models.**

lines 378-380: need to reference section 2.3.1 to tell reader that the appropriate GCM simulations will be available.
**Done.**

lines 381-389: Uncertainty in the driving land use/cover data poses the same challenge for comparison to observations. This needs to be acknowledged here as well, and I would expect it to be discussed more thoroughly in the LUH2 paper. Related to the uncertainty in the driving land use/cover data is the remaining uncertainty due to the translation of land use to land cover, which includes differences between land cover classes and plant functional types, the changes in non-forest land cover (which are not harmonized), and differences between the definition of forest in the LUH2 data and in the models, and how different models will implement the forest cover changes (e.g. prognostic vs. non-prognostic biogeography models). I don't think it is possible to explore the model sensitivity to land use/cover uncertainty in cmip6, but this exploration should be noted as a target for future cmips and land mips, with the potential for using additional land use/cover data sets to drive the models.

**This topic will be covered in more detail in the LUH2 paper, but the point is well taken.  We have elected to include a couple of additional experiments in the set of land-only simulations with alternative plausible land-use reconstructions.  These simulations will allow for sensitivity analysis of the impact of different land-use histories.**

phase 2 experiments

line 395: "Historical" seems like an extra word here

**It's actually not an extra word.  Somewhat awkwardly, the CMIP6 Historical simulation is not part of the DECK or a satellite MIP so it is in it's own category.  We reworded slightly to try to make sentence clearer.**

line 399: describe all the relevant aspects of the cmip6 historical concentration-driven simulation. for example, what land use/management processes are included?

**As above, we can't specify this. Each modeling group will make their own decisions about what to include in their CMIP6 historical simulations. All we can do is ask for information about what aspects of land use were included.**

line 416: you may want to move your parenthetical note about ssp scenarios from line 420 to here. You should also include the relevant details of the parent sims here. e.g., what land use and land management activities are active.

**Done. Same as above with respect to what land use and land management activities are active. It is up to each group to decide.**

line 422: land management isn't isolated in these experiments. the changes will be a combination of differences in land use, cover, and management (same issue in figure 3 caption). there may be individual pixels that can be extracted that have only land management/use/cover differences, but there will also be dependencies on the surrounding land what the total effects are for a given pixel. At the subgrid level this may work out for the crop data, but only if there are comparable crop areas between sims within the given pixels and only the management options are different (e.g. irrigation and fertilizer).

**We reword to downplay how much we can do with respect to providing input on land policy, but we still keep this sentiment since we think that, although the results will not be directly relevant to policy (i.e., not policy prescriptive), with careful analysis one should be able at least infer the impacts of different land use and land management decisions on future climate.**

line 434: again, land management isn't isolated in these sims. and there will effects of surrounding land on a given pixel.

**See above.**

land use metrics and analysis plans

line 491: paired simulation analyses means that you need to ensure that your main control sims (which are shared with other mips) are well described in this paper as well the lumip specific ones, so that your lumip experiments are clear.

**As we have noted before, we feel that the descriptions for the LUMIP experiments are as clear as we can make them and that we should not describe the experiments from the other MIPs in detail since it is those MIPs that have the responsibility to fully define them. By pointing the user to the relevant simulation in the other MIPs, we can ensure that groups complete the simulation as requested by that MIP. If we reproduce the description in the LUMIP paper, there is a significant probability that the descriptions will be out of sync and/or incorrect.**

lines 527-531: redundant sentences

**Fixed. Thanks.**

line 533: rfmip - another acronym needing expanding

**Done.**

lines 535-540: i suggest briefly describing the rfmip land experiment and how it complements lumip to make this paragraph more relevant.

**Looking at the simulations in RFMIP, it is actually not easy to simply explain the experiment that isolates land-use ERF because it actually involves three experiments.  Providing enough detail for a reader to be able to understand what was done would make the paragraph too cumbersome, in our opinion.  The main point is the result and readers can refer to either the RFMIP paper or the Andrews et al. paper for details.**

lines 543-558: this is a good idea, and differences in land coupling strength among models may (or may not) also help identify where land use/cover/management may be different among models.

**We think you mean where land use /cover/management impacts may be different among models.  We make that point in the text.**

lines 592-594: This is a good idea, but I think that the forest and non-forest areas need to be separated out to replace the primary/secondary land category, to the extent possible (i expect that not all requested variables are kept track of at the forest/nonforest level, but some of them, such as carbon, are kept track of at the pft level in some models). for variables and models that do not distinguish between forest and nonforest, the primary/secondary value can be placed in one category with a flag in the other signifying that land cover is not segregated. This may not be practically feasible due to how the models store and write outputs, however, so it is something to consider for future comparisons, and maybe with more land cover types distinguished from each other.

**We acknowledge that there are many ways to try to condense the vast amount of data that land models can potentially produce.  After many discussions, we elected to go with the four listed land use types.  We believe that this set of land use types will produce the most information for the smallest amount of additional data.  Each model is likely going to need to aggregate their subgrid output in different ways to conform with the request. We accept that the request for archival of subgrid land use information is to a certain degree experimental and we anticipate that there will be problems encountered along the way.  One of our goals, though, is to push the community to at least start thinking about archiving and utilizing subgrid data.  We believe that our request will do that and that the experience gained through the process will provide the basis for modifications for future MIPs whether they be CMIP or other.**

lines 601-604: figures 6 and 7 don't seem to help much here, as they are not complete and clear about the variables (e.g., only biogeochemistry is shown, and fig 7 shows processes rather than variables). A table of all the requested variables, with the subgrid ones noted, would be more useful. please provide a link or a supplemental table of the full list of variables requested.

**We removed Figure 6 and have redrawn Figure 7. In addition, we have added a list of variables, with the caveat that the list is subject to change.  "A list of requested land-use tile variables is shown in Table 5.  However, this list is subject to change.  Modelers should refer to the CMIP6 output request documents for the final variable list.  "**

lines 651-654: please reference figure 9 if you want to include it.

**Done.**

line 674: for the future runs, land management isn't isolated (or will be extremely difficult to isolate, even at the subgrid level). you can get information about this from the historical land-only experiments, however.

**We agree, and have reworded to note that these experiments will be useful to provide preliminary assessment of how land use and land management could be utilized to mitigate climate.**

line 677: not sure what you mean by "relative to fossil fuel emissions." It seems that the experiments are designed to quantify the effects of lulcc, in a more absolute sense, which can then be compared to the total emissions effects. I don't see quantification of fossil fuel effects only, nor outputs that would be lulcc effects relative to fossil fuel effects.

**The impact of LULCC change emissions relative to fossil fuel emissions should be able to be inferred through the no LULCC experiments, but we take your point that we do not have the experiments to explicitly assess this. We change to "effects on climate of LULCC relative to all forcings."**

Tables and Figures

Table 1 please include other simulation required for the experiment it should be more clear that this is a tier 1 experiment

**We refer to the piControl experiment in the comments.**

Table 2 it should be more clear which tier the experiments are in, and this should be noted in the same column for all. I suggest stating the tier at the beginning of each description or notes column, for each experiment. Or adding a narrow "tier" column on the right, with the appropriate number indicated. land-hist needs to be clearly marked as a sim that is shared by another mip. land-crop-nomanage: is all crop area constrained to 1850? so this is like a constant crop sim, and the pasture area and harvest can change over time? can irrigation amounts change? what about fertilization area?

**We prefer not to add a column. Priority is listed in Table caption. Added text indicating that land-hist is shared. Crop area is transient (now noted). This simulation in combination with lnd-crop-noIrrig and lnd-crop-noFert helps isolate the impact of crop management through irrigation and fertilization. Irrigation amounts are not specified by LUH2, only irrigated-equipped area.**

what is a "prognostic crop model" and how does it differ from what is used in the control sim? The description needs to be more complete as to what is different from the land-hist sim land-crop-nofert: i suggest two more sims to ask questions about the effects of changing area vs changing amounts: one with constant area and changing rate, and one with constant rate and changing area. land-netTrans: unclear what it means to maintain gross transitions in excess of net transitions also, the degree to which spatially gross transitions are included at coarser resolution depends on the upscaling process; the finer grid cells can be summed to get a net change for a coarser grid cell.

**We agree that the term prognostic crop model is confusing and have removed the term prognostic. We only mean to distinguish between the treatment of croplands as unmanaged grasslands versus with crops with some form of management, especially including explicit planting and harvesting. The suggestion for additional simulations that would more effectively isolate specific aspects of fertilization are good ones, but we have elected not to include them because the list of experiments is already long. Our hope/intention is that once these experiments are underway, individual modeling groups or several groups together can elect to conduct additional factorial simulations to probe even further where appropriate for their model. At some stage, insightful additional simulations could potentially be added to the overall protocol through the forum/email list.**

**Regarding the land-netTrans, we have reconsidered the land-netTrans experiment and decided that it would be clearer if we simply specify this as a no shifting cultivation simulation. Both the language and the concept is now clearer. We have added a figure that explains what we mean by shifting cultivation.**

Table 3 the tier of each simulation needs to be clearly marked. i suggest adding rows for the control cmip6 sim, and the tier 2 and 3 ensemble members. tier 3 needs to be explained in the text. see comments above.

**We now more clearly demarcate the Tiers for each experiment.**

Table 4 This does not seem necessary, as this information, plus more, is directly available in the text.

**We prefer to retain Table 4, even though it is relatively simple, for readers who want to quickly scan the document to see how we have defined the land use tiles.**

Figure 3 Why note only 1 of the 3 additional lumip sims in the caption? note all or none.
maybe state that the brown text are the additional sims.

**Text has been removed.  We think it is clear that the brown text represents the LUMIP sims so have not added anything.**

Figure 4 I would classify wood harvest as land use, with various types of silviculture/
harvest (e.g. tree selection, clear cut, plantation, coppice) as land management.
see the definitions you invoke in section 1.3.

**Corrected.**

Figure 5 This figure and its caption is not consistent with the section on net lulcc emissions.

**We have rewritten the caption and the net LULCC emission section to make it clearer and remove inconsistencies.**

LULCC emissions are also "seen" by vegetation in prescribed transient CO2 sims also because the historical atmosphere data include all emissions (LULCC occurred historically) and the IAM projected CO2 emissions include their respective estimates of LULCC emissions. Furthermore, figure 5c also has LASC, even with constant CO2, because different land covers have different potential rates of carbon uptake.

**We have revised the text in both the main paper and the figure caption to improve clarity of the discussion here:  "The loss of additional sink capacity (LASC) is a factor when environmental conditions change transiently, which is the case when historical $CO_2$ concentrations, which implicitly include increases in $CO_2$ due to fossil-fuel burning (FFB) and LULCC, are prescribed from observations. Prognostic LULCC emissions are directly "seen" by the terrestrial vegetation (natural and anthropogenic) only in the ESM setup, where $CO_2$  is interactive. In this case, a fraction of the LULCC emissions is taken up again by the vegetation ("land-use carbon feedback")."**

**Agreed that technically the amount of CO2 that could be taken up can change even in a constant CO2 run, but the concept of a sink typically refers to a situation where CO2 is evolving.  In any case, we believe that the main point of this section is to note that care needs to be taken when assessing LULCC carbon fluxes across different model configurations.  Discussion within the research community is ongoing about this topic and LUMIP will certainly be involved in those discussions.**

Figure 9 not referenced by text

**Fixed.**
1. Page 2, Abstract, 1st line – Missing word. I think "changes" should go just after "large" and before "to the Earth surface"

**Corrected.**

2. Page 2, lines 29-30 – Should mention need for documentation of what the groups did to run the experiment. These details are at least as important to trying to follow the experimental design.

**Added request for documentation to the abstract.**

3.page 3, line 59 – 40% of radiative forcing – What is time period? When to when: : :

**Modified text to say "…accounts for ~45% of the total historic (1850 to 2010) changes in radiative forcing (Ward et al. 2014)."**

4. Page 6, line 142 – "industrial roundwood" – What is this? Please define.

**We amended the sentence to: "… fuelwood and industrial roundwood (i.e., timber that is cut for uses other than for fuel)."**

5. Page 8, section 2 – I think there should be multiple mentions of the need for documentation of what was done and how by the modeling groups. Each group's land model is quite different from the others. The details will be very important if we are going to be able to figure out the results after the experiments are completed.

**Agreed.  We have attempted to make this clearer and will be communicating with all the groups explicitly and frequently to make this request.  Google group is already setup for communication.**

6. Page 10 – Several references to Figure X – line 276, 282, 299. Please insert correct figure number. 6B. Page 11, lines 314, 333
**Corrected.**

7. Page 13, top – This discussion is confusing to me. Cleanly discuss the various types of errors: model, forcing, observations.

**We have rewritten the paragraph to improve clarity.**